# Small Cajal body-associated RNA 2 (scaRNA2) regulates DNA repair pathway choice by inhibiting DNA-PK

Sofie Bergstrand [1,4], Eleanor M. O'Brien [2,4], Christos Coucoravas[2], Dominika Hrossova[2], Dimitra Peirasmaki[2], Sandro Schmidli [2], Soniya Dhanjal[2], Chiara Pederiva [2], Lee Siggens[1], Oliver Mortusewicz [3], Julienne J. O'Rourke [2] & Marianne Farnebo [1,2✉]

Evidence that long non-coding RNAs (lncRNAs) participate in DNA repair is accumulating, however, whether they can control DNA repair pathway choice is unknown. Here we show that the small Cajal body-specific RNA 2 (scaRNA2) can promote HR by inhibiting DNA-dependent protein kinase (DNA-PK) and, thereby, NHEJ. By binding to the catalytic subunit of DNA-PK (DNA-PKcs), scaRNA2 weakens its interaction with the Ku70/80 subunits, as well as with the LINP1 lncRNA, thereby preventing catalytic activation of the enzyme. Inhibition of DNA-PK by scaRNA2 stimulates DNA end resection by the MRN/CtIP complex, activation of ATM at DNA lesions and subsequent repair by HR. ScaRNA2 is regulated in turn by WRAP53β, which binds this RNA, sequestering it away from DNA-PKcs and allowing NHEJ to proceed. These findings reveal that RNA-dependent control of DNA-PK catalytic activity is involved in regulating whether the cell utilizes NHEJ or HR.

[1] Department of Biosciences and Nutrition, Neo, Karolinska Institutet, Stockholm, Sweden. [2] Department of Cell and Molecular biology, Biomedicum, Karolinska Institutet, Stockholm, Sweden. [3] Department of Oncology and Pathology, SciLife, Karolinska Institutet, Stockholm, Sweden. [4]These authors contributed equally: Sofie Bergstrand, Eleanor M. O'Brien. ✉email: marianne.farnebo@ki.se

DNA double-strand breaks, which threaten the stability of the genome, can be repaired by two main pathways, i.e., non-homologous end-joining (NHEJ) or homologous recombination (HR). Via NHEJ, the predominant pathway, the broken ends are simply rapidly rejoined. It involves binding of DNA ends by the heterodimer Ku70/Ku80, which together with DNA-PKcs (DNA-dependent protein kinase catalytic subunit) forms the holoenzyme that facilitates recruitment of the downstream factors XRCC4, XLF, and DNA ligase IV which mediate rejoining. In contrast, HR is relatively slow and, since it utilizes an undamaged sister chromatid to promote repair, is largely restricted to the S and G2 phases of the cell cycle[1].

HR is initiated by resection of the broken ends by the MRN (MRE11–RAD50–NBS1) complex and the CtIP endonuclease. Subsequently, the single-stranded DNA is bound by RPA, which is then replaced by the recombinase RAD51 to form a filament that initiates the search for homology. BRCA1 participates in RAD51-mediated homologous pairing and, at the same time, acts upstream to promote DNA resection[1]. In addition, the MRN complex recruits and activates ATM, which phosphorylates a number of proteins, thereby promoting HR and activating a variety of complementary cellular responses[2,3]. The efficiency of NHEJ together with the extensive expression of Ku70/Ku80 and their subsequent rapid binding to most free DNA ends suggests that in mammals NHEJ is the default repair pathway[4]. Consequently, when accurate repair by HR is required, the factors that attenuate NHEJ play crucial roles.

Evidence for the involvement of RNA in DNA repair is accumulating. For example, RNA transcribed from damaged DNA can promote repair either by hybridizing with the complementary DNA[5–9] or, after being converted into small double-stranded RNAs by DICER and DROSHA, stimulating the accumulation of key repair factors[10–13]. Moreover, long non-coding (lnc) RNAs, including LINP1, BGL3 and DDSR1, can serve as scaffolds for the formation of repair complexes or to prevent unscheduled binding of repair proteins[14–16]. Still, it appears likely that RNA regulates DNA repair in other ways as well. In this context, it is noteworthy that small nucleolar RNAs (snoRNAs) assembled in the nucleolus where they regulate ribosomal RNAs can stimulate the activities of the repair enzymes DNA-PK and PARP1 in the absence of DNA damage[17,18]. Although to date this stimulation has only been shown to take place within the nucleolus and influence ribosome biogenesis, these or related RNAs may regulate repair enzymes in other contexts as well.

RNAs that are highly similar in structure and function to snoRNAs are the so-called small Cajal body-associated RNAs (scaRNAs), which assemble in Cajal bodies to modify spliceosomal small nuclear (sn)RNAs[19–21], or to elongate telomeres (the telomerase RNA is a scaRNA). Antisense elements within the scaRNAs allow them to hybridize to their targets and subsequently promote site-specific pseudouridylation or methylation of these targets by associated proteins. The interaction between scaRNAs and these proteins is guided by specific sequence motifs within the scaRNA: either the box H/ACA domain which forms a hairpin-hinge-hairpin-tail structure that associates with the pseudouridylation complex (i.e., dyskerin, NOP10, NHP2 and GAR1); or the box C/D domain that forms a kink-turn structure that is bound by the methylation complex (i.e., fibrillarin, 15.5 kDa/NHP2L1, NOP56 and NOP58). In contrast to snoRNAs, which usually contain only one of these boxes, scaRNAs often contain two C/D or H/ACA domains or one of each (and are therefore referred to as mixed-domain scaRNAs)[19,22].

In addition to these motifs, scaRNAs contain a Cajal body (CAB) box which directs them to this organelle and the sequence of which differs between scaRNAs containing H/ACA (ugAG) or C/D boxes (cgaGUUAnUg in *Drosophila* or GU/UG-rich wobble stem in humans[23–25]. ScaRNAs form ribonucleoprotein (RNP) complexes with Cajal body proteins, such as WRAP53β (WD40 encoding RNA Antisense to p53), coilin and TDP-43, which regulates the localization and processing of these RNAs[23,24,26–29]. WRAP53β is essential for correct localization of scaRNAs to the Cajal body[23,24,26,27] and abnormalities in this protein have been linked to several genetic disorders[30–36]. Interestingly, WRAP53β is recruited rapidly to DNA double-strand breaks to regulate their repair[37–40], indicating an involvement of scaRNAs in this same process.

Here, we show that scaRNA2 function as an endogenous inhibitor of the DNA-PK enzyme. This RNA localizes to DNA breaks, where it supports initiation of HR by the MRN complex. Mechanistically, scaRNA2 achieves this by inhibiting assembly of the DNA-PK holoenzyme, thus blocking NHEJ and promoting HR. This investigation highlights the involvement of RNA in controlling the catalytic activity of DNA-PK during DNA repair, and moreover, in the regulation of repair pathway choice at double-strand breaks.

## Results

**scaRNA2 associates with chromatin as part of the DNA damage response.** We reasoned that greater insight into the organization of scaRNA genes in humans might reveal novel functions of these molecules, similar to the manner in which snoRNAs were found to be involved in ribosome biogenesis[41–43]. Our mapping of 29 scaRNAs revealed that 27 of these are encoded from introns of genes that also encode either proteins (24), or non-coding RNAs (3). Only scaRNA2 and TERC are transcribed from their own promoters (Supplementary Table 1). Notably, the majority of proteins encoded from scaRNA-host genes are involved in DNA repair and/or chromatin organization, indicating that scaRNAs may have similar functions. To explore their potential involvement in DNA repair, we chose to focus on scaRNA2, since it associates extensively with WRAP53β[23,26,27,30], shows altered processing following DNA damage[44] and its overexpression has been connected with cancer progression and chemoresistance[45]. Moreover, since scaRNA2 is transcribed independently[46], the risk of indirect effects due to co-depletion of factors encoded by the host gene upon knockdown is minimal.

ScaRNA2 is 420 nucleotides long in humans, evolutionarily conserved (present in dogs, mice, rats, frogs, fish, ciona, fly and plant) and proposed to guide 2'-O-methylation of residues 25 and 61 in U2 snRNA (which is why scaRNA2 is also known as mgU2-25/61) (Fig. 1a and Supplementary Fig. 1a)[25,47]. This RNA is transcribed by RNA polymerase II as a non-spliced unit with a 5' cap, but no poly A-tail[46,47]. Its GU-rich sequence in the 5'-region is important for stability and localization to Cajal bodies, while the two subsequent C/D domains are involved in the modification of snRNA (Fig. 1a and Supplementary Fig. 1a)[24,27,47]. A shorter isoform of scaRNA2 predicted to obstruct a snoRNA involved in methylation of ribosomal RNA has also been identified[47,48]. Applying subcellular fractionation, we found that full-length scaRNA2 is recovered in the nuclear subfraction, as expected, but that for unknown reasons more than 90% is associated with chromatin (Fig. 1b and Supplementary Fig. 1b, c). Damaging DNA with ionizing radiation (IR) altered neither this distribution nor the total level of scaRNA2 (Fig. 1b and Supplementary Fig. 1d).

Single-molecule (sm) RNA FISH demonstrated that scaRNA2, while distributed weakly throughout the nucleoplasm, is strongly enriched in Cajal bodies, as reported previously[27] and confirmed here by co-staining for the protein coilin, a marker for these nuclear organelles (Fig. 1c). ScaRNA2 overexpressed without or with an MS2 stem loop (scaRNA2-MS2) was localized in a similar manner

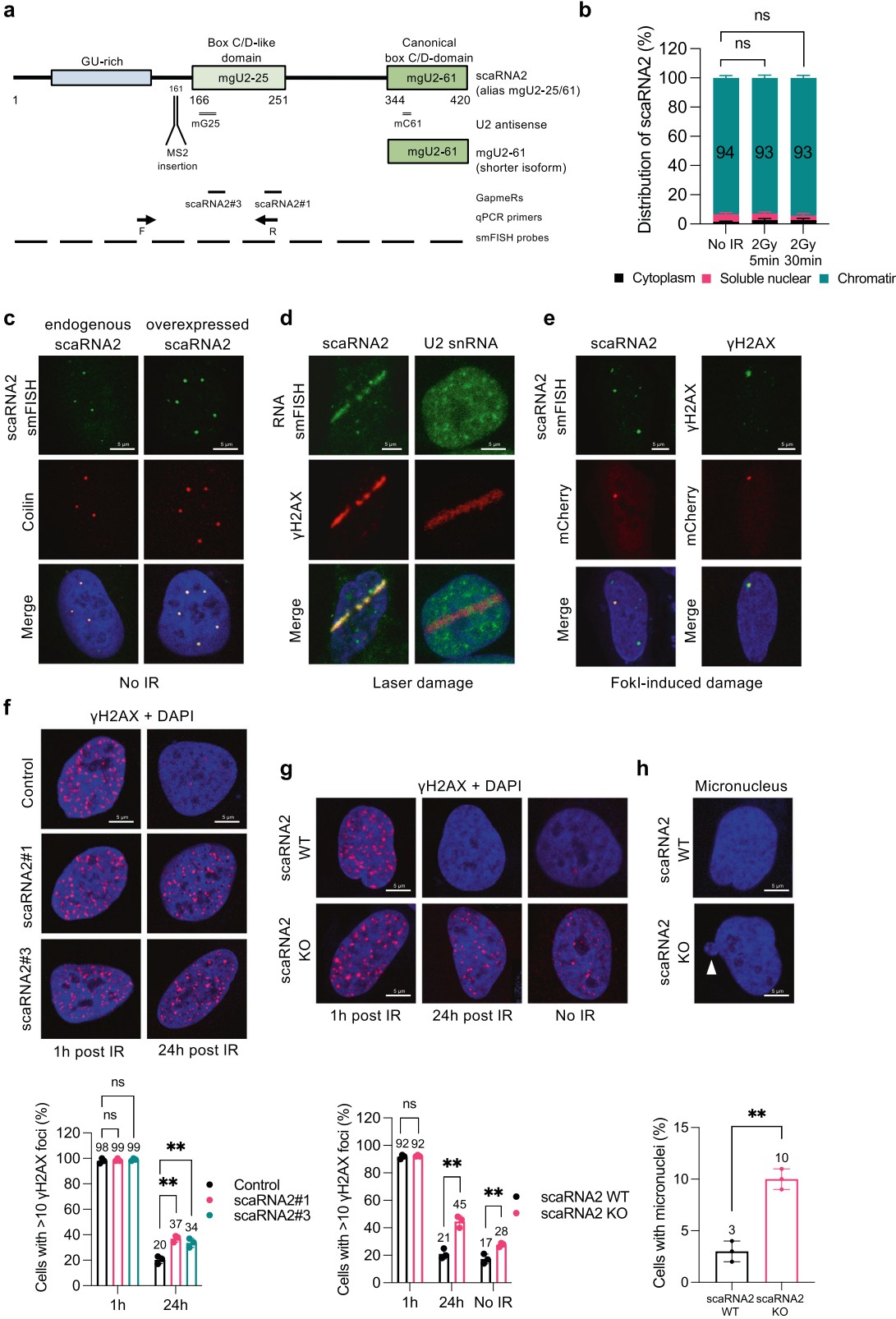

(Fig. 1c and Supplementary Fig. 2a). The specificity of the scaRNA2 FISH staining was confirmed by showing that its signal was reduced by knockdown employing antisense LNA oligonucleotides (GapmeRs) (Supplementary Fig. 2b). The expression of other scaRNAs was not changed upon depletion of scaRNA2 (Supplementary Fig. 2c).

Interestingly, following induction of DNA damage by laser micro-irradiation, scaRNA2 localized to sites of DNA breaks (Fig. 1d and Supplementary Fig. 2d), whereas this was not the case for the U2 snRNA (Fig. 1d). Moreover, when double-strand breaks were generated with the FokI endonuclease, scaRNA2 was also observed at these DNA damage sites (Fig. 1e and

**Fig. 1 scaRNA2 localizes at DNA lesions and loss of this RNA leads to accumulation of DNA double-strand breaks and genomic instability. a** Schematic structure of scaRNA2, illustrating the GU-rich region, C/D domains and U2 snRNA antisense regions. Binding sites for smFISH probes, qPCR primers, GapmeRs and the MS2 loop insertion site are also shown. **b** Sub-cellular distribution of scaRNA2 in untreated or irradiated U2OS cells as determined by qPCR. The values shown are means ± SD, $n = 3$. Unless otherwise indicated, all $n = 3$ refer to three biologically independent experiments. Ns (not significant) as determined by one-way ANOVA and two-sided Dunnett's multiple comparisons test. **c** smRNA FISH of scaRNA2 and immunostaining of the Cajal body marker coilin in U2OS cells expressing scaRNA2 endogenously or overexpressed for 24 h ($n = 3$). Nuclei were stained with DAPI in all immunofluorescence experiments. **d** smRNA FISH of scaRNA2 and immunostaining of the DNA damage marker γH2AX in laser micro-irradiated (5 min recovery) U2OS cells overexpressing scaRNA2 for 24 h ($n = 3$). smRNA FISH for U2 snRNA was performed under the same conditions but without overexpression of scaRNA2. **e** smFISH of scaRNA2 in U2OS FokI cells transfected with a scaRNA2 plasmid for 24 h and treated with Shield and 4-OHT for an additional 4 h ($n = 3$). Immunostaining of γH2AX was performed under the same conditions but without overexpression of scaRNA2. **f** Immunostaining of γH2AX in irradiated (2 Gy, 1 or 24 h recovery) U2OS cells depleted or not of scaRNA2 for 48 h. The graph below shows the percentage of 100–200 cells (means ± SD, $n = 3$) whose nuclei contained > 10 γH2AX foci, **$p < 0.01$, ns (not significant) as determined by one-way ANOVA and two-sided Dunnett's multiple comparisons test. The images depict a representative cell exhibiting the potential change in phenotype. **g** Immunostaining of γH2AX in untreated or irradiated (2 Gy, 1 or 24 h recovery) U2OS scaRNA2 WT or KO cells. The graph below shows the percentage of 100–200 cells (means ± SD, $n = 3$) whose nuclei contained >10 γH2AX foci, **$p < 0.01$, ns (not significant) as determined by unpaired two-tailed t-test. **h** DAPI staining in U2OS scaRNA2 WT or KO cells. The graph below shows the percentage of 100–200 cells (means ± SD, $n = 3$) that contained a micronucleus (indicated by a white arrowhead in the representative image), **$p < 0.01$ as determined by unpaired two-tailed t-test. Source data are provided as a Source Data file.

Supplementary Fig. 2e, f). Localization of scaRNA2 at DNA double-strand breaks was only possible when this RNA was overexpressed and even then, only in relatively few cells. This may be due to the presence of only small numbers of scaRNA2 molecules at these lesions and/or the transient localization of scaRNA2 at these sites. Nevertheless, our findings demonstrate translocation of scaRNA2 to bona fide double-strand breaks.

To explore the potential role of scaRNA2 in the DNA damage response, the clearance of γH2AX repair foci after IR was monitored in cells containing or lacking scaRNA2. This showed a rapid induction of γH2AX foci following IR treatment in both control and scaRNA2-depleted cells (Fig. 1f). However, whereas in control cells these foci were resolved within 24 h, significant numbers remained even after 24 h in the cells lacking scaRNA2, indicating aberrant repair (Fig. 1f and Supplementary Fig. 2g). In addition, accumulation of residual γH2AX foci in both U2OS and MCF7 cells was also observed following stable knock-out (KO) of scaRNA2 employing CRISPR/Cas9 (Fig. 1g and Supplementary Fig. 2h, i). In these stable KO cells, particularly after long-term culture, we also observed accumulation of spontaneous γH2AX foci, as well as the presence of micronuclei, both signs of genomic instability (Fig. 1g, h). Together, these findings reveal that scaRNA2 localizes to DNA lesions and plays an important role in the repair of DNA double-strand breaks.

**scaRNA2 promotes recruitment of HR factors to double-strand breaks and subsequent repair by HR.** To explore the involvement of scaRNA2 in the repair of DNA double-strand breaks in greater detail, we examined its potential influence on the accumulation of repair factors. Since this RNA interacts with WRAP53β, which is known to promote ubiquitination of damaged chromatin by the RNF8/RNF168 E3 ligases and subsequent recruitment of 53BP1, BRCA1 and RAD51[37,38,49], we examined these repair factors first. Following exposure of cells depleted of scaRNA2 (either transiently or stably knocked out) to irradiation (Fig. 2a, b) or the FokI endonuclease (Fig. 2c), foci containing WRAP53β, conjugated ubiquitin, and 53BP1 still formed at the double-strand breaks. At the same time, accumulation of several key factors in the HR pathway, including BRCA1, RAD51 and RPA2, was lowered. Accumulation of RAD51 and BRCA1 at repair foci could be restored fully by re-introducing scaRNA2 (with or without an MS2 tag) into KO cells (Fig. 2d and Supplementary Fig. 3a), arguing against the involvement of indirect effects. Thus, a deficiency in scaRNA2 impairs assembly of HR factors at DNA double-strand breaks, without influencing the ubiquitin signaling at these sites.

To examine whether this impairment due to scaRNA2 deficiency actually impairs repair of DNA breaks by HR, we examined the efficiency of this repair pathway in U2OS cells stably transfected with a reporter construct containing a direct repeated (DR)-GFP sequence[50]. Treatment of these cells with GapmeRs for scaRNA2 lowered the efficiency of HR by 30–40%, compared to the 70% reduction upon depletion of RAD51 (the positive control) (Fig. 2e and Supplementary Fig. 3b–d). Conversely, following overexpression of scaRNA2, HR efficiency was around 130% higher than in cells transfected with an empty vector (Fig. 2f and Supplementary Fig. 3e, f), highlighting scaRNA2 as a potential rate-limiting factor in the HR response. The lack of any significant alteration in the cell cycle following scaRNA2 depletion (Supplementary Fig. 3g–i) excluded possible indirect effects of this nature. Since scaRNA2 has been proposed to methylate U2 snRNA, which is involved in pre-mRNA splicing, we also excluded this potential indirect effect by demonstrating that the expression of repair factors, as well as splicing efficiency were unaltered in cells depleted of scaRNA2 (Supplementary Fig. 4a–d). Together, these findings demonstrate that scaRNA2 participates in the recruitment of factors required for HR to double-strand breaks, thereby promoting subsequent repair by this pathway.

**scaRNA2 promotes accumulation of the MRN complex and activation of ATM at DNA breaks.** We next wished to analyze the potential involvement of scaRNA2 in the initial steps of HR, including recognition of DNA damage and DNA end resection by the MRN (MRE11, RAD50 and NBS1) complex in combination with the CtIP endonuclease. For this purpose, we damaged cellular DNA with FokI, which facilitates detection of these factors at double-strand breaks. This approach revealed that accumulation of CtIP and MRN components at FokI-induced breaks in cells depleted of scaRNA2 was reduced to a similar extent as accumulation of RPA2 and BRCA1 (Fig. 3a).

Following recognition of DNA damage, the MRN complex recruits and activates ATM, which in turn converts H2AX into its phosphorylated form, γH2AX. In cells lacking scaRNA2, both activation of ATM (recognized through phosphorylation at S1981) and formation of γH2AX at FokI-induced double-strand breaks were reduced (Fig. 3b), indicative of a defect in the early stages of HR. Since the overall accumulation of ATM at these breaks was unaltered (Fig. 3b), scaRNA2 appears to attenuate the activation, rather than the recruitment of this factor. Accordingly, following irradiation of cells lacking scaRNA2, both the number and intensity of pATM foci were lowered (Fig. 3c, d), an effect

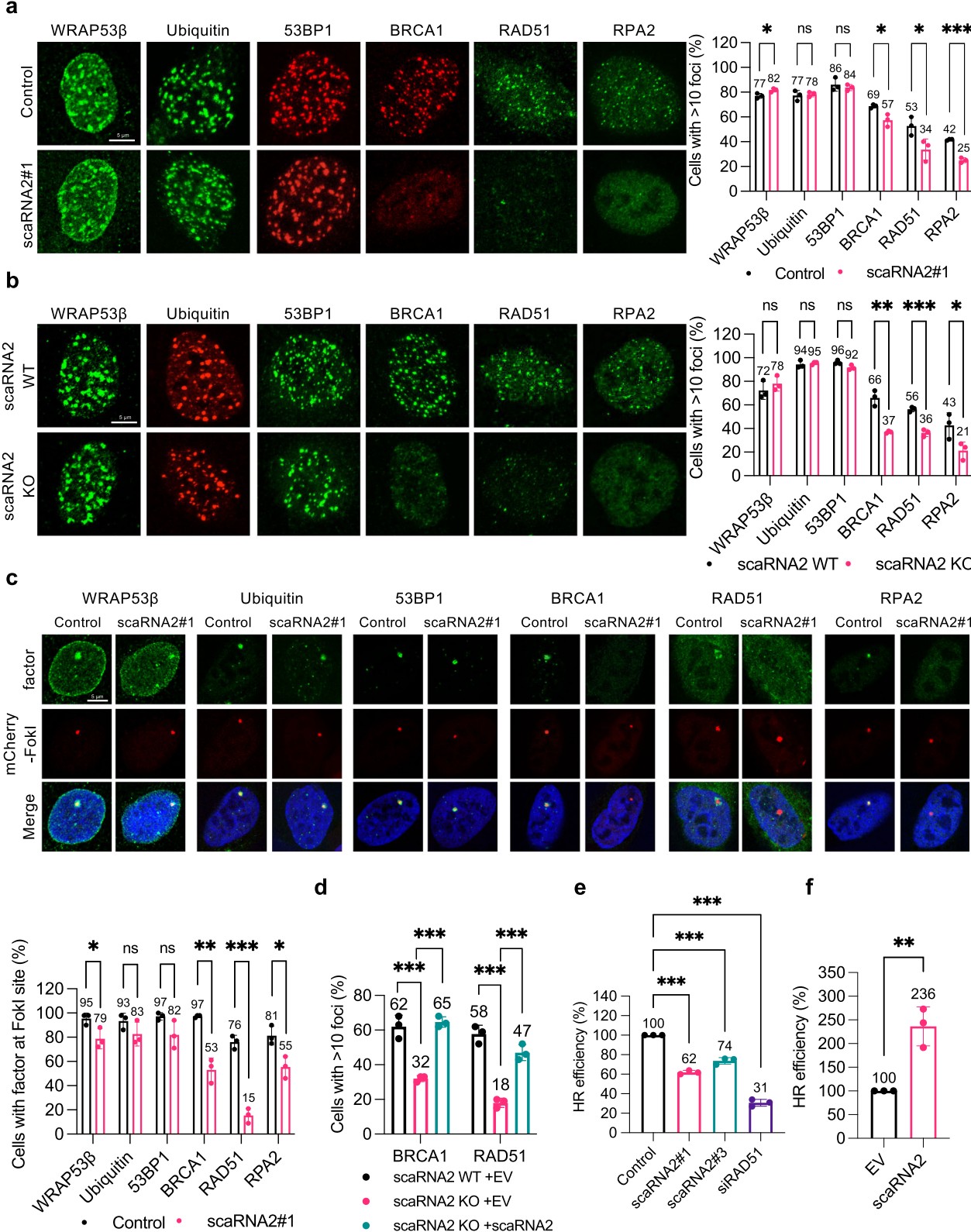

that persisted for a longer period (Supplementary Fig. 5a). Although the numbers of γH2AX IR-induced foci in control and scaRNA2-depleted cells were similar, the intensity of these foci was lower in the absence of scaRNA2 (Fig. 3c, d). Again, potential indirect effects due to altered expression of any of the proteins involved could be excluded (Supplementary Fig. 4a). We conclude that scaRNA2 is involved in very early stages of the HR cascade,

where it is required for accumulation of the MRN complex and activation of ATM.

**scaRNA2 associates with DNA-PK and restricts NHEJ.** To explore the mechanism through which scaRNA2 influences HR, we examined its potential binding to components of the MRN complex employing native RIP, but observed no interactions

**Fig. 2 Loss of scaRNA2 impairs recruitment of HR proteins to double-strand breaks and subsequent repair by HR. a** Immunostaining of the repair proteins indicated in U2OS cells transfected with GapmeRs for 48 h and then irradiated (2 Gy, 1 h recovery). The graph shows the percentage of 100–200 cells (means ± SD, $n = 3$) whose nuclei contained >10 IR-induced foci, $*p < 0.05$, $***p < 0.001$, ns (not significant) as determined by unpaired two-tailed t-test. The images depict a representative cell exhibiting the potential change in phenotype. Note: Although the results for only one GapmeR are presented, at least two different GapmeRs targeting scaRNA2 were employed in all cases with similar results. **b** Immunostaining of the repair proteins indicated in irradiated (2 Gy, 1 h recovery) U2OS scaRNA2 WT or KO cells. The graph shows quantification performed in the same manner as described above, $*p < 0.05$, $**p < 0.01$, $***p < 0.001$, ns (not significant) as determined by unpaired two-tailed t-test. **c** Immunostaining of the repair proteins indicated in U2OS FokI cells transfected with GapmeRs for 48 h and treated with Shield and 4-OHT for another 4 h. The graph below shows the percentage of 50 cells (means ± SD, $n = 3$) in which the protein indicated co-localized with the FokI site, $*p < 0.05$, $**p < 0.01$, $***p < 0.001$, ns (not significant) as determined by unpaired two-tailed t-test. **d** smFISH of scaRNA2 and immunostaining of BRCA1 or RAD51 in U2OS scaRNA2 WT or KO cells transfected with an empty vector (EV) or a scaRNA2 plasmid for 24 h, and then irradiated (2 Gy, 1 h recovery). The graph shows the percentage of 100–200 cells (means ± SD $n = 3$) whose nuclei contained >10 IR-induced foci, $***p < 0.001$ as determined by one-way ANOVA and two-sided Dunnett's multiple comparisons test. In the case of scaRNA2-transfected cells, only those in which the overexpressed scaRNA2 was localizing predominantly in Cajal bodies were counted. **e** The efficiency of HR following treatment of DR-GFP U2OS cells with the GapmeRs or siRNA indicated for 48 h. GFP expression was analyzed by flow cytometry. The values shown are means ± SD $n = 3$, $***p < 0.001$ as determined by one-way ANOVA and two-sided Dunnett's multiple comparisons test. **f** The efficiency of HR following overexpression of scaRNA2 or an empty vector (EV) for 48 h. GFP expression was analyzed by flow cytometry. The values shown are means ± SD $n = 3$, $**p < 0.01$ as determined by unpaired two-tailed t-test. Source data are provided as a Source Data file.

(Supplementary Fig. 6a). Interestingly, similar experiments with upstream factors involved in the competing NHEJ pathway revealed that scaRNA2 co-precipitates with Ku70 and, even more extensively, with the catalytic subunit of DNA-PK (DNA-PKcs) (Fig. 4a and Supplementary Fig. 6b). Employing crosslinking and immunoprecipitation (CLIP), scaRNA2 was shown to bind DNA-PKcs directly, whereas the binding of Ku70/80 to scaRNA2 was primarily indirect (Fig. 4b). The association between scaRNA2 and Ku70 was elevated in cells depleted of DNA-PKcs (Fig. 4c and Supplementary Fig. 6c), indicating that DNA-PKcs and Ku70 compete for binding to scaRNA2.

Since activation of the NHEJ pathway by DNA-PK suppresses HR[51–54], it appeared possible that the impaired HR in cells lacking scaRNA2 is a consequence of aberrant NHEJ. To facilitate visualization of many of the proteins that participate in NHEJ at DNA lesions[55–57], we damaged cellular DNA with FokI, which generates multiple breaks within the LacO cassette, and, in addition, removed background signaling by extracting soluble proteins with detergent. This allowed us to visualize DNA-PKcs, Ku70, Ku80, XRCC4, and DNA ligase IV at breaks, but only in a small proportion of the cells (Fig. 4d). Notably, following depletion of scaRNA2 more of these proteins were observed at these breaks and in a larger proportion of the cells (Fig. 4d). Consistent with the hyperactivation of the NHEJ pathway, the efficiency of this repair process was enhanced in cells lacking scaRNA2 (Fig. 4e), an effect that could be reversed through overexpression of scaRNA2 (Supplementary Fig. 6d), indicating that scaRNA2 suppresses NHEJ repair.

To confirm this conclusion, we treated cells with NU7441, an inhibitor of DNA-PK (DNA-PKi). As expected, this prevented autophosphorylation of DNA-PKcs at S2056, a widely accepted indicator of DNA-PK activation in cells (Supplementary Figs. 6e and 7c). Notably, in the case of scaRNA2-depleted cells, treatment with DNA-PKi restored accumulation of RAD51 at sites of DNA damage (Fig. 4f), suggesting that in the absence of scaRNA2, HR is impaired due to excessive activation of DNA-PK and NHEJ. Altogether, these results indicate that scaRNA2 promotes HR by repressing NHEJ.

**scaRNA2 inhibits the catalytic activity of DNA-PK.** Our findings that scaRNA2 binds to DNA-PKcs and represses NHEJ led to the hypothesis that scaRNA2 regulates the catalytic activity of DNA-PK, which we tested employing an in vitro kinase assay in which DNA-PK autophosphorylates DNA-PKcs. As expected, recombinant holo-DNA-PK (consisting of DNA-PKcs and Ku70/80) was activated and DNA-PKcs autophosphorylated at S2056 and T2609 upon addition of double-stranded DNA and ATP

(Fig. 5a). Strikingly, addition of in vitro transcribed scaRNA2 at the same time as the DNA eliminated autophosphorylation of DNA-PK completely, while DNA-PK remained inactive in the presence of scaRNA2 and absence of DNA (Fig. 5a). Ten nanograms of scaRNA2 proved sufficient to inhibit activation of DNA-PK by 10 ng linearized plasmid (Fig. 5b). This inhibition was not prevented by increasing the concentration of Ku70/80 (Supplementary Fig. 7a), further supporting the conclusion that DNA-PKcs, but not Ku70/80, is the target of scaRNA2.

To identify the key region of scaRNA2 involved in this effect, the ability of deletion variants to prevent autophosphorylation of DNA-PK in vitro was examined. A deletion mutant lacking a significant stretch of GU dinucleotides retained this ability (Fig. 5c), indicating that this region is not important. The C/D domains of scaRNA2 (mgU2-25 and mgU2-61) could independently reduce autophosphorylation of DNA-PK (Fig. 5c) and as these domains can each form a kink-turn structure, these results indicate that the structure, rather than the sequence of scaRNA2 mediates inhibition of DNA-PK autophosphorylation in vitro. Indeed, scaRNA5 and scaRNA7, each of which contains one of these domains, also caused a decline in this autophosphorylation to a varying extent (Supplementary Fig. 7b). We conclude that the kink-turn structure of scaRNA2 is important for the inhibition of DNA-PK catalytic activity.

We then characterized this inhibition in greater detail in scaRNA2-depleted cells. Since in cells S2056 is autophosphorylated, whereas T2609 is phosphorylated primarily by ATM and ATR[58–60], phosphorylation of S2056 was utilized as an indicator of DNA-PK activation. As would be expected from the inhibition of DNA-PK by scaRNA2, in cells depleted of scaRNA2, autophosphorylation of DNA-PK was elevated — both basal pDNA-PK levels and following exposure to irradiation, observed in cell lysates (Fig. 5d) and at double-strand breaks (Fig. 5e). The specificity of the pDNA-PK immunostainings was verified by the reduction in signal caused by treatment with DNA-PKi (Supplementary Fig. 7c). Furthermore, ATM phosphorylation in lysates of irradiated scaRNA2-depleted cells was lowered (Fig. 5d), a known effect of DNA-PK activation[51] and also in agreement with the impairment of ATM signaling at DNA breaks in cells lacking scaRNA2 described above.

To obtain additional insight into how DNA-PK can be activated in cells containing scaRNA2, we characterized potential changes in the interaction between DNA-PKcs and scaRNA2 in response to irradiation. Notably, while the extent of interaction between scaRNA2 and the total pool of DNA-PKcs remained unchanged, binding of scaRNA2 to activated DNA-PK (autophosphorylated at S2056) declined around 70% within minutes after

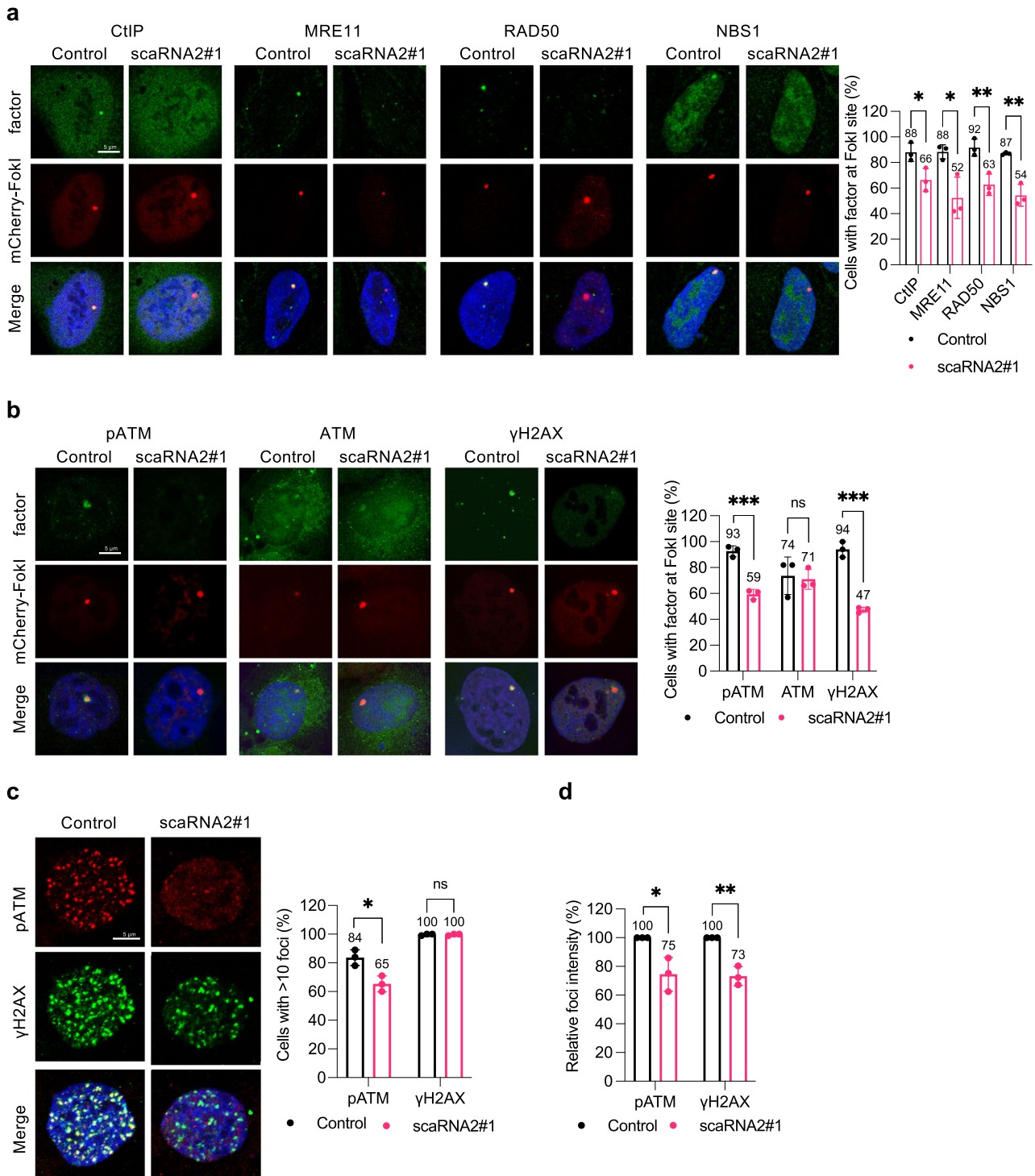

**Fig. 3 Loss of scaRNA2 reduces ATM activation and recruitment of the MRN complex at DNA double-strand breaks. a, b** Immunostaining of the repair proteins indicated in U2OS FokI cells transfected with GapmeRs for 48 h and treated with Shield and 4-OHT for 4 h. The graph shows the percentage of 50 cells (means ± SD $n = 3$) in which the protein indicated co-localized at the FokI site, *$p < 0.05$, **$p < 0.01$, ***$p < 0.001$, ns (not significant) as determined by unpaired two-tailed t-test. The images depict a representative cell exhibiting the potential change in phenotype. **c** Immunostaining of the repair proteins indicated in U2OS cells transfected with GapmeRs for 48 h and then irradiated (2 Gy, 1 h recovery). The graph illustrates the percentage of 100–200 cells (means ± SD $n = 3$) whose nuclei contained > 10 IR-induced foci, *$p < 0.05$, ns (not significant) as determined by unpaired two-tailed t-test. **d** The relative intensity (control set to 100%) of pATM and γH2AX IR-induced foci in the same cells depicted in Fig. 3c, as assessed using CellProfiler. The values shown are means ± SD $n = 3$, *$p < 0.05$, **$p < 0.01$ as determined by unpaired two-tailed t-test. Source data are provided as a Source Data file.

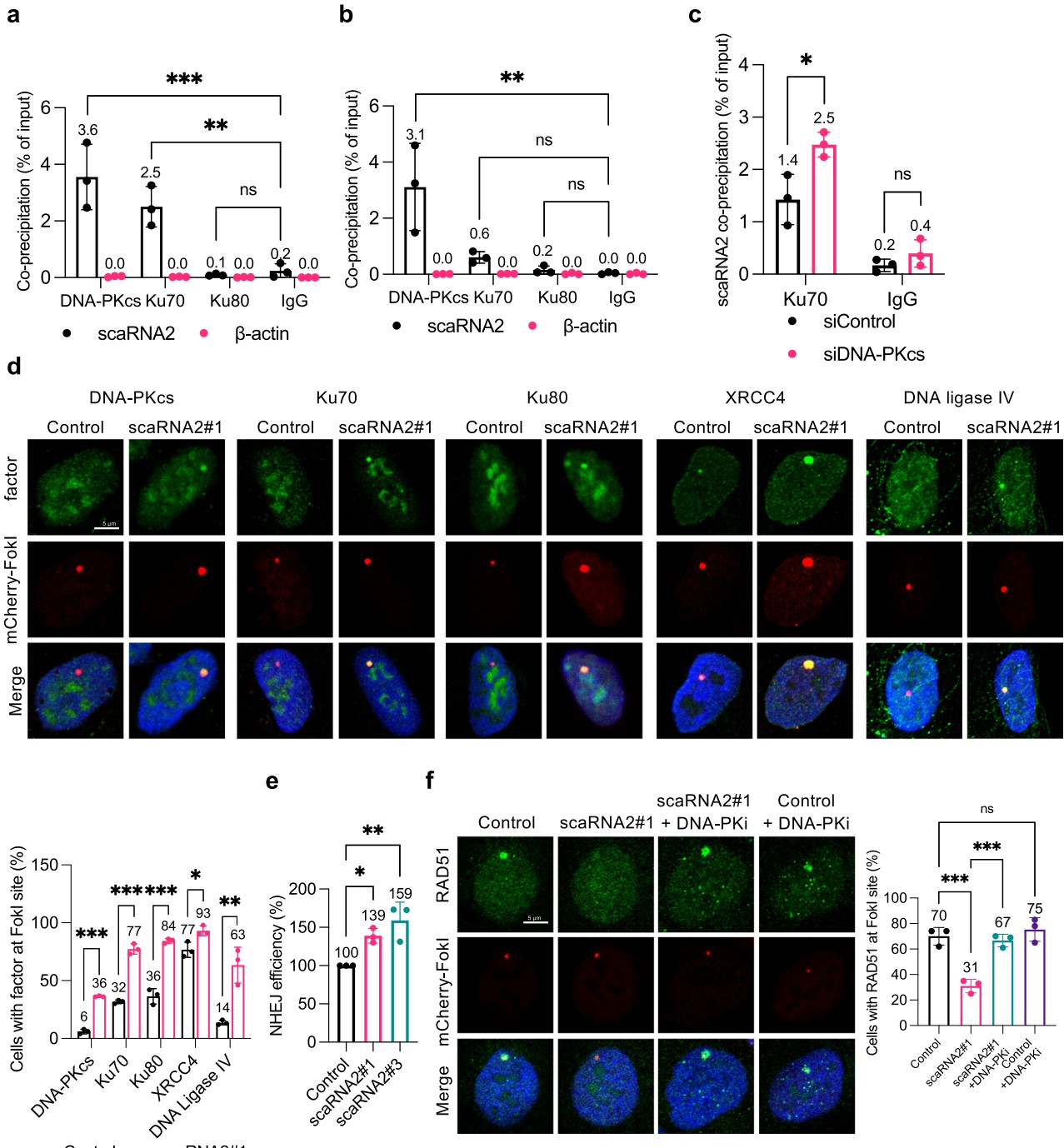

irradiation (Fig. 5f and Supplementary Fig. 7d), an effect which persisted for at least 24 h (Supplementary Fig. 7e). Together, these findings demonstrate that scaRNA2 inhibits activation of DNA-PK both in vitro and in cells and is released from DNA-PKcs during its activation by DNA damage, observations consistent with a role of scaRNA2 as an inhibitor of this enzyme.

**scaRNA2 inhibits DNA-PK by blocking its assembly and through competition with LINP1 RNA.** The kinase activity of DNA-PKcs is stimulated upon binding to Ku70/80-DNA[61–63], which raises the possibility that scaRNA2 influences this binding. Immunoprecipitation of Ku80 in scaRNA2-depleted cells revealed enhanced binding of this factor to DNA-PKcs, while binding between Ku80 and Ku70 remained unchanged (Fig. 6a). The

enhanced interaction of Ku proteins (and in particular Ku80) with DNA-PKcs was found to be even more pronounced upon immunoprecipitation of autophosphorylated DNA-PK from cells lacking scaRNA2 (Fig. 6b), both with transient and stable scaRNA2 knockdown (Fig. 6b and Supplementary Fig. 8a). Thus, in the absence of scaRNA2, the components of DNA-PK associate more strongly, suggesting that abnormal assembly of these components might be the underlying cause of defective recruitment of HR proteins to DNA breaks following knockdown of scaRNA2. Indeed, depletion of DNA-PKcs restored accumulation of RAD51 at DNA breaks in cells lacking scaRNA2 completely (Fig. 6c and Supplementary Fig. 8b–d), providing additional support for the conclusion that scaRNA2 promotes HR by blocking the assembly of DNA-PK.

**Fig. 4 scaRNA2 binds to DNA-PKcs, thereby suppressing NHEJ. a, b** Co-immunoprecipitation of RNA (RIP) with DNA-PKcs, Ku70, Ku80 or IgG (negative control) from U2OS cells using (**a**) native or (**b**) UV-crosslinking (CLIP) conditions. The graphs show the amount of co-precipitated RNA as percentage of input (means ± SD $n = 3$) measured by qPCR, **$p < 0.01$, ***$p < 0.001$, ns (not significant) as determined by one-way ANOVA and two-sided Dunnett's multiple comparisons test. Representative blots depicting the immunoprecipitation of DNA-PKcs, Ku70, Ku80, and IgG antibodies are presented in Supplementary Fig. 6b. **c** Native RIP of Ku70 from U2OS cells depleted or not of DNA-PKcs for 48 h, as indicated. The graph shows the amount of co-precipitated RNA as percentage of input (means ± SD $n = 3$) as determined by qPCR, *$p < 0.05$, ns (not significant) as determined by unpaired two-tailed t-test. **d** Immunostaining of the repair proteins indicated in U2OS FokI cells transfected with GapmeRs for 48 h and treated with Shield and 4-OHT for an additional 4 h. The graph below shows the percentage of 50 cells (means ± SD $n = 3$) in which the protein indicated co-localized at the FokI site, *$p < 0.05$, **$p < 0.01$, ***$p < 0.001$ as determined by unpaired two-tailed t-test. The images depict a representative cell exhibiting the potential change in phenotype. **e** The efficiency of NHEJ following treatment of U2OS EJ5-GFP NHEJ cells with the GapmeRs indicated for 48 h. GFP expression was analyzed by flow cytometry. The values shown are means ± SD $n = 3$, *$p < 0.05$, **$p < 0.01$ as determined by one-way ANOVA and two-sided Dunnett's multiple comparisons test. **f** Immunostaining of RAD51 in U2OS FokI cells transfected with GapmeRs for 48 h and treated with Shield and 4-OHT for another 4 h. In addition, where indicated, the cells were exposed to 10 μM DNA-PK inhibitor for 1 h prior to induction of double-strand breaks. The graph shows the percentage of 50 cells (mean ± SD $n = 3$) in which RAD51 co-localized at the FokI site, ***$p < 0.001$, ns (not significant) as determined by one-way ANOVA and two-sided Šidák's multiple comparisons test. Source data are provided as a Source Data file.

Important for the assembly of DNA-PK is the binding of Ku80 to a region near the PQR autophosphorylation cluster (including the S2056 site) of DNA-PKcs[64,65]. We investigated whether scaRNA2 targets this region of DNA-PKcs utilizing the 18-2 monoclonal antibody that binds amino acids 2001–2025 of DNA-PKcs, thereby inhibiting its catalytic activity[66,67]. This antibody failed to precipitate scaRNA2 bound to DNA-PKcs, whereas immunoprecipitation of DNA-PKcs with antibodies targeting other sites co-precipitated approximately 2–3% of the total cellular scaRNA2 (Fig. 4a, b, and Supplementary Fig. 8e, f). On the basis of these results and also considering that DNA can prevent binding of 18-2 to DNA-PKcs[66], as confirmed here and also observed to be the case with RNA (Supplementary Fig. 8g, h), we propose that scaRNA2 binds to residues 2001–2025 of DNA-PKcs or, alternatively, alters the confirmation of DNA-PKcs in a manner that masks this epitope. Thus, binding of scaRNA2 to DNA-PKcs prevents the interaction of DNA-PKcs with Ku80 that is required for enzyme assembly and kinase activation.

Recently, interaction between DNA-PKcs and Ku80 was shown to be stimulated by the LINP1 scaffold RNA[15]. Interestingly, knockdown of LINP1 in cells lacking scaRNA2 attenuated hyperphosphorylation of DNA-PK and restored the accumulation of RAD51 at DNA break sites (Fig. 6d, e and Supplementary Fig. 9a–d). Moreover, binding of LINP1 to pDNA-PKcs was enhanced in cells depleted either transiently or stably of scaRNA2 (Fig. 6f and Supplementary Fig. 9e), which indicates that LINP1 is a component of the abnormal DNA-PK complex formed in the absence of scaRNA2.

In contrast, depletion of LINP1 elevated the binding of scaRNA2 to pDNA-PKcs (Fig. 6g) and lessened the association between DNA-PKcs and Ku70/80 (Fig. 6h). This is consistent with scaRNA2 blocking DNA-PK assembly and also reveals a competition between scaRNA2 and LINP1 in DNA-PK activation. In the DNA-PK kinase assay, LINP1 stimulated autophosphorylation of DNA-PK (Fig. 6i), although less efficient than double-stranded DNA (Fig. 6i). This effect of LINP1 was attenuated in the presence of scaRNA2 (Fig. 6i), demonstrating that scaRNA2 can block both DNA- and RNA-dependent activation of DNA-PK. Thus, LINP1 and scaRNA2 play opposing roles in connection with DNA-PK activation, with inhibition by scaRNA2 appearing to be more potent than stimulation by LINP1.

To gain more detailed insight into the regulation of scaRNA2 and LINP1, their expression during the various phases of the cell cycle was examined. The level of scaRNA2 remained relatively constant, with somewhat higher expression during the G2 phase, when repair by HR is favored (Fig. 6j and Supplementary Fig. 9f). In contrast, expression of LINP1 was lowest during G2 and highest during the G1 phase (Fig. 6j), when NHEJ is dominant.

No changes in the expression of LINP1 or its localization to chromatin occurred following DNA damage (Supplementary Fig. 9g) or depletion of scaRNA2 (Supplementary Fig. 9h). In summary, this reveals that an interplay between scaRNA2 and LINP1 is involved in the regulation of DNA-PK activity in response to DNA damage, as well as providing evidence that by reducing assembly of the DNA-PK complex, scaRNA2 prevents activation of DNA-PK.

**Binding to and inhibition of DNA-PKcs by scaRNA2 is regulated by WRAP53β.** Next, the potential involvement of WRAP53β, coilin, TDP-43, and fibrillarin, all of which bind to scaRNA2, in inhibition of DNA-PK by scaRNA2 was explored. Utilizing CLIP, we first demonstrated direct interaction of scaRNA2 with all of these proteins, with none of them binding LINP1 (Fig. 7a and Supplementary Fig. 10a). Co-precipitation of scaRNA2 was most pronounced in the presence of WRAP53β and coilin and these proteins also bound extensively to other scaRNAs containing either a C/D and/or H/ACA box (Supplementary Fig. 10b). Subsequent immunoprecipitation of the individual components of DNA-PK revealed interaction with WRAP53β and coilin, but not with TDP-43 or fibrillarin (Fig. 7b), indicating that a complex containing WRAP53β, coilin and scaRNA2 is involved in regulating DNA-PK activation.

To examine this possibility further, addition of WRAP53β in combination with scaRNA2 to the DNA-PK kinase assay was found to prevent inhibition of DNA-PK activity by scaRNA2, with DNA-PK becoming autophosphorylated to the same extent as in control cells (Fig. 7c). This effect of WRAP53β was independent of coilin, TDP-43 and fibrillarin (Supplementary Fig. 10c, d). Moreover, in cells depleted of WRAP53β, association between scaRNA2 and DNA-PKcs was more extensive (Fig. 7d); whereas this was not the case following depletion of coilin, TDP-43, or fibrillarin, which, if anything, reduced this association (Supplementary Fig. 10e). These findings indicate that WRAP53β and DNA-PK compete for binding to scaRNA2 and that a preferential binding of scaRNA2 to WRAP53β can prevent inhibition of DNA-PK.

Subsequently, we found that autophosphorylation of DNA-PK was lowered in cells lacking WRAP53β, both with and without irradiation (Fig. 7e and Supplementary Fig. 10f, g). Depletion of scaRNA2 in cells lacking WRAP53β restored hyperphosphorylation of DNA-PK (Fig. 7e), which is consistent with the idea that scaRNA2 is responsible for inactivation of DNA-PK following depletion of WRAP53β. Consequently, in the absence of WRAP53β, NHEJ repair was impaired (Fig. 7f)[37]. Together, these observations demonstrate that WRAP53β can decrease the binding to and inhibition of DNA-PKcs by scaRNA2. Thus, in the absence of WRAP53β, more scaRNA2 binds to DNA-PK, preventing its activation and NHEJ repair.

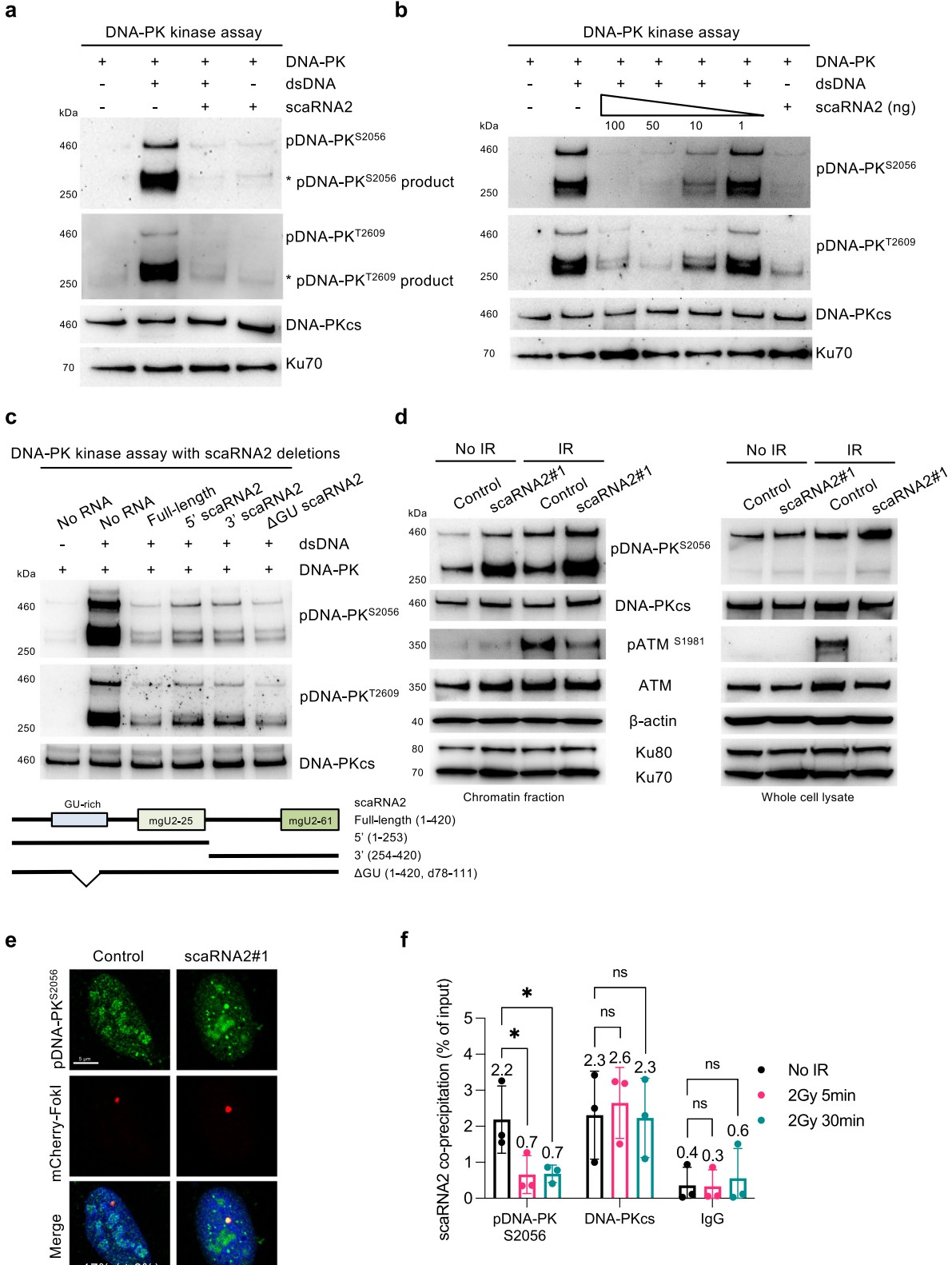

## Discussion

The current view is that when DNA-PK is assembled at DNA breaks, it undergoes a conformational change that activates its catalytic activity[64,65,68,69]. In this context, endogenous inhibitors of DNA-PK activation remain largely unexplored. Here, we identify scaRNA2 as such an inhibitor, which upon binding to DNA-PKcs weakens its interaction with Ku70/80, thereby preventing assembly and activation. Consequently, NHEJ is suppressed and DNA repair by HR promoted. At the same time, WRAP53β can bind and sequester scaRNA2, allowing DNA-PKcs activation and NHEJ repair (Fig. 7g).

In our search for novel functions of members of the scaRNA family, we discovered that many of these RNAs are coded from introns of genes that also encode proteins involved in DNA

**Fig. 5 scaRNA2 inhibits the catalytic activity of DNA-PK. a** In vitro assay of DNA-PK kinase activity employing recombinant human DNA-PK (25–30 U), double-stranded (ds) DNA (10 ng) or in vitro transcribed scaRNA2 (100 ng), followed by western blotting of the proteins indicated. The blots shown are representative of >3 independent experiments. The asterisks indicate degradation products and/or a smaller isoform of phosphorylated DNA-PKcs, as determined by MS analysis of the DNA-PK complex[17] and detection by all anti-pDNA-PK$^{S2056}$ antibodies employed in the experiments performed. This band is most distinct following autophosphorylation of recombinant DNA-PK in vitro, as well as in chromatin fractions prepared from cells. **b** In vitro assay of DNA-PK kinase activity in the presence of 1–100 ng scaRNA2, followed by western blotting. The blots shown are representative of two independent experiments. **c** In vitro assay of DNA-PK kinase activity in the presence of variants of scaRNA2 containing different deletions, followed by western blotting. The blots shown are representative of >3 independent experiments. **d** U2OS cells were transfected with GapmeRs for 48 h, with or without subsequent exposure to irradiation (6 Gy), and 1 h later the proteins of whole cells or their chromatin fractions were extracted for western blotting. The blots shown are representative of >3 independent experiments. **e** Immunostaining of pDNA-PK$^{S2056}$ in U2OS FokI cells transfected with GapmeRs for 48 h and treated with Shield and 4-OHT for an additional 4 h. The white numbers represent the percentage of 50 cells (means ± SD $n = 3$) in which pDNA-PK$^{S2056}$ co-localized at the FokI site, *$p < 0.05$ as determined by unpaired two-tailed t-test. **f** Native RIP with pDNA-PK$^{S2056}$, total DNA-PKcs or IgG from U2OS cells treated with irradiation (2 Gy) and left to recover for 5 or 30 min. The graph shows the amount of co-precipitated RNA as percentage of input (means ± SD $n = 3$) measured by qPCR, *$p < 0.05$, ns (not significant) as determined by one-way ANOVA and two-sided Dunnett's multiple comparisons test. Representative blots depicting immunoprecipitation of DNA-PKcs and pDNA-PK$^{S2056}$ are presented in Supplementary Fig. 7d. Source data are provided as a Source Data file.

repair. Moreover, more than 90% of scaRNA2 was found to be associated with chromatin and to accumulate at DNA double-strand breaks, which is indicative of a functional role in DNA repair. Visualization of scaRNA2 at DNA lesions was only possible following overexpression of this RNA and, even then, only in a fraction of the cells. This might reflect the association of a relatively small number of scaRNA2 units with each double-strand break, as is known to be the case for many NHEJ factors[55–57], or alternatively that scaRNA2 is present at these sites for a relatively short period of time or during specific cell cycle phases. Hopefully, more sensitive detection of scaRNA2 will help unravel the mechanism by which the presence of this RNA at sites of damage is controlled.

We show that scaRNA2 is required for the initial steps of HR, stimulating the resection of DNA ends by the MRN/CtIP complex, as well as activation of ATM at DNA lesions. However, stimulation of HR by scaRNA2 is an indirect consequence of inactivation of DNA-PK and thereby the competing NHEJ pathway. The choice of NHEJ or HR for DNA repair is regulated in a highly complex manner and includes cross-talk between DNA-PKcs and ATM, mediated in part by their phosphorylation of one another[51,56,58,70]. Our finding that DNA-PK activity is enhanced and ATM signaling reduced in the absence of scaRNA2 is consistent with such reciprocal interaction and supports a model where presence of scaRNA2 counteract inhibition of ATM and of MRN/CtIP-mediated end resection by DNA-PK, favoring HR. Thus, scaRNA2 can be added to the list of repair factors involved in the cellular choice of pathway by which double-strand breaks are repaired.

DNA-PKcs, a huge protein kinase consisting of 4128 amino acids and with a molecular weight of approximately 469 kD, contains a long N-terminus composed largely of HEAT repeats (residues 1-2801) and a C-terminal region (residues 2801-4128) that harbors the FAT, kinase and FATC domains[64,65,69]. Here, we show that scaRNA2 inhibits catalytic activation of DNA-PKcs in a manner affecting autophosphorylation at both S2056 and T2609. Thus, rather than obstructing phosphorylation at a particular site, scaRNA2 blocks the general activation of DNA-PKcs, perhaps by disrupting formation of the DNA-PK complex.

The catalytic activity of DNA-PKcs is regulated allosterically upon its binding to Ku70/80/DNA, with the binding of Ku80 near the PQR autophosphorylation cluster (including S2056) playing a key role in initiation of complex formation[64,65]. Factors binding to and blocking this region of DNA-PKcs, including BRCA1 (which targets residues 1878-2182)[71] and the monoclonal antibody 18-2 (residues 2001–2025)[66,67], prevent activation of DNA-PK. Our

present results show that in the absence of scaRNA2, association between DNA-PKcs and Ku70/80 is enhanced and DNA-PK is hyperactivated, which indicates that scaRNA2 also interferes with formation of the enzyme complex. Moreover, precipitation of DNA-PKcs with the 18-2 antibody does not co-precipitate scaRNA2, further indicating that scaRNA2 and 18-2 may bind to the same site, thereby preventing the interaction with Ku80 required for enzyme assembly and kinase activation.

Interestingly, the abnormal DNA-PK complex formed in the absence of scaRNA2 also contains elevated amounts of LINP1, a lncRNA that can facilitate interaction between Ku80 and DNA-PKcs by binding both simultaneously, as shown both here and previously[15]. Although the site on DNA-PKcs at which LINP1 binds is presently unknown, a recent structural study revealed that LINP1 binds the central β-barrel between Ku70 and Ku80, thereby promoting formation of Ku multimers and the initial synaptic event for NHEJ at DNA ends[72]. While depletion of scaRNA2 enhanced binding of LINP1 to pDNA-PKcs, knockdown of LINP1 in turn elevated the binding of scaRNA2 to pDNA-PKcs, suggesting that scaRNA2 and LINP1 compete for binding to DNA-PKcs. Our finding that LINP1 is part of the hyperactive DNA-PK complex formed in the absence of scaRNA2 indicates that this lncRNA may aid the activation of DNA-PK. Indeed, knockdown of LINP1 in cells lacking scaRNA2 attenuated hyperphosphorylation of DNA-PK and restored HR signaling at DNA lesions. This observation, together with our finding that inhibition of DNA-PK in these same cells also restored HR signaling, constitute convincing evidence that the repression of HR caused by lack of scaRNA2 is due to enhanced complex formation and subsequent hyperactivation of DNA-PK. Thus, there is an intricate interplay between scaRNA2 and LINP1 in connection with the regulation of DNA-PK in response to DNA damage.

Here, we also show that WRAP53β attenuates the inhibitory effect of scaRNA2 on the DNA-PK complex and that WRAP53β thereby acts as a positive regulator of DNA-PK catalytic activity. This is achieved by preferential binding of scaRNA2 to WRAP53β, which reduces the interaction between scaRNA2 and DNA-PKcs. This function of WRAP53β appeared to be independent of coilin, TDP-43, and fibrillarin. At the same time, both coilin and TDP-43 appear to play roles in NHEJ repair[73–75], so it is still possible that their binding to scaRNA2 is of significance in this connection. Interestingly, stimulation of telomerase activity by WRAP53β also involves its binding to a scaRNA, TERC, which triggers a conformational change within TERC[76]. Thus, WRAP53β may alter the structure of scaRNA2 in a manner that lowers its ability to bind to and inhibit DNA-PKcs.

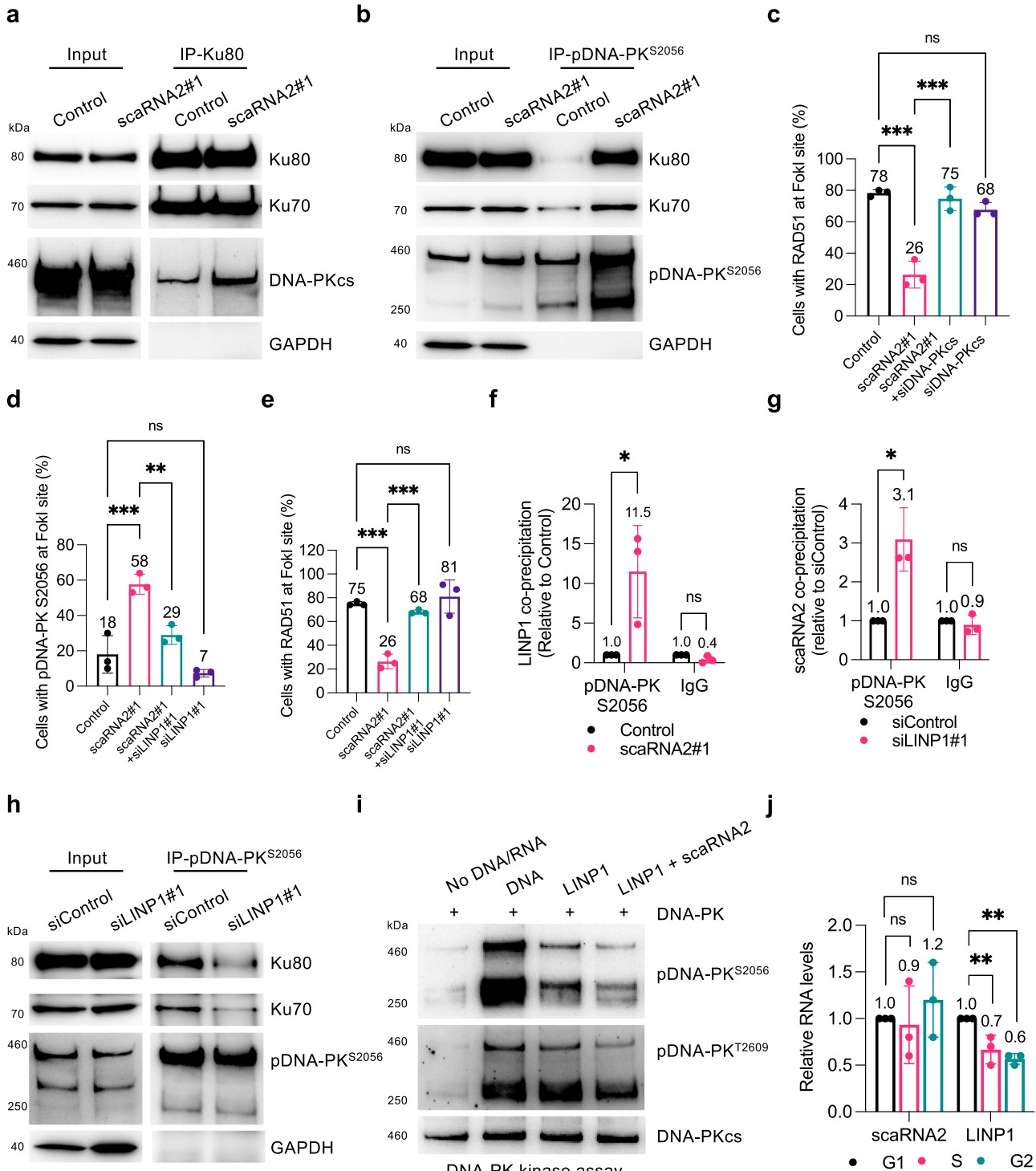

In the present context it is noteworthy that mutations in WRAP53β, some of which prevent binding to scaRNA2, can result in dyskeratosis congenita and the Hoyeraal–Hreidarsson syndrome[30] and that these patients suffer from many of the key symptoms associated with DNA-PK deficiency, including bone marrow failure, severe immunodeficiency, hematopoietic defects, and growth retardation[17,77–80].

In summary, the current investigation unveils scaRNA2 as an endogenous inhibitor of DNA-PK and demonstrates that this RNA, through the restriction of DNA-PK activation, is essential for HR-mediated repair of DNA double-strand breaks.

## Methods

**Cells and culture conditions**. The cells and culture conditions used in this study are shown in Supplementary Table 2. Dulbecco's modified Eagle medium (DMEM) and DMEM plus GlutaMAX™ supplement were supplied by Gibco, Thermo Fisher Scientific. All culture media were supplemented with 10% fetal bovine serum (FBS) (HyClone) and 1% penicillin/streptomycin (Gibco, Thermo Fisher Scientific), and cells were grown and maintained at 37 °C under 5% $CO_2$ in humidified incubators. DNA-PK inhibitor NU7441 was obtained from Tocris Bioscience and used at a concentration of 10 μM.

**Sub-cellular fractionation**. Cells were lysed in Buffer A (5 mM PIPES, 85 mM KCl, 0.5% NP40) for 10 min on ice, followed by centrifugation at 2500 × g for 5 min at

**Fig. 6 scaRNA2 inhibits DNA-PK by both blocking its assembly and through competition with LINP1. a, b** Immunoprecipitation (IP) of (**a**) Ku80 or (**b**) pDNA-PK$^{S2056}$ from irradiated (6 Gy IR, 30 min recovery) U2OS cells transiently depleted of scaRNA2 as indicated, followed by western blotting. The blots shown are representative of 3 independent experiments. 6 Gy was applied to achieve more extensive interaction/co-precipitation between Ku70/80 and DNA-PKcs. **c** Immunostaining of RAD51 in U2OS FokI cells depleted of scaRNA2 and/or DNA-PKcs for 48 h and treated with Shield and 4-OHT for an additional 4 h. The graph shows the percentage of 50 cells (means ± SD $n = 3$) in which RAD51 co-localized at the FokI site, ***$p < 0.001$, ns (not significant) as determined by one-way ANOVA and two-sided Šidák's multiple comparisons test. Representative images depicting staining of RAD51 in these cells are presented in Supplementary Fig. 8b. **d, e** Immunostaining of pDNA-PK$^{S2056}$ (**d**) or RAD51 (**e**) in U2OS FokI cells depleted of scaRNA2 and/ or LINP1 for 48 h and treated with Shield and 4-OHT for an additional 4 h. The graph shows quantification performed in the same manner as described above, **$p < 0.01$, ***$p < 0.001$, ns (not significant) as determined by one-way ANOVA and two-sided Šidák's multiple comparisons test. Representative images depicting staining of pDNA-PK$^{S2056}$ or RAD51 in these cells are presented in Supplementary Fig. 9a, b. **f** Native RIP of pDNA-PK$^{S2056}$ or IgG from irradiated (6 Gy, 30 min recovery) U2OS cells transiently depleted of scaRNA2 as indicated. The graph shows the amount of co-precipitated RNA (means ± SD $n = 3$) as determined by qPCR and presented as fold-enrichment relative to Control, *$p < 0.05$, ns (not significant) as determined by unpaired two-tailed t-test. **g** Native RIP of pDNA-PK$^{S2056}$ or IgG from U2OS cells transiently depleted of LINP1 as indicated. The graph shows quantification performed in the same manner as described above, *$p < 0.05$, ns (not significant) as determined by unpaired two-tailed t-test. **h** IP of pDNA-PK$^{S2056}$ from irradiated U2OS cells (6 Gy, 30 min recovery) transiently depleted of LINP1 as indicated, followed by western blotting. The blots shown are representative of 3 independent experiments. **i** In vitro assay of DNA-PK kinase activity in the presence of dsDNA, LINP1 and/or scaRNA2 as indicated, followed by western blotting. The blots shown are representative of >3 independent experiments. **j** qPCR analysis of LINP1 and scaRNA2 expression in G1, S, or G2-phase of synchronized U2OS cells. The levels of scaRNA2 were normalized to those of β-actin and the values shown are relative to the G1 sample (the 0 h time point) (means ± SD $n = 3$), **$p < 0.01$, ns (not significant) as determined by one-way ANOVA and two-sided Dunnett's multiple comparisons test. Source data are provided as a Source Data file.

4 °C, and the resulting supernatant stored as the cytoplasmic fraction. The remaining nuclear pellet was washed once in Buffer A without NP40, then resuspended in NP40 buffer (137 mM NaCl, 50 mM Tris (pH 8.0), and 1% NP40), passed through a 25 G needle 4 times, centrifuged at 1500 × g for 10 min at 4 °C, and the resulting supernatant stored as the soluble nuclear fraction. The remaining chromatin pellet was resuspended in NP40 buffer containing high salt (400 mM NaCl, 50 mM Tris (pH 8.0), and 1% NP40) and DNA sheared by 5 cycles of sonication (30 sec ON/30 sec OFF, low intensity) in a Bioruptor (Diagenode). Following centrifugation at 16000 × g for 10 min at 4 °C, the resulting supernatant was saved as the chromatin fraction. RNA was extracted from each individual fraction using TRIzol® (Life Technologies, ThermoFisher Scientific) and the RNeasy Mini Kit (Qiagen) in accordance with the manufacturer's instructions.

**RT-qPCR.** cDNA was generated with the SuperScript IV reverse transcriptase (Invitrogen, ThermoFisher Scientific), random hexamer primers (ThermoFisher Scientific), 10 nM dNTPs Mix (ThermoFisher Scientific) and RNaseOUT (Invitrogen, ThermoFisher Scientific) in accordance with the manufacturers' instructions. The levels of specific species of RNA were determined by RT-qPCR in a 7500 Fast Real-Time PCR System (Applied Biosystems, ThermoFisher Scientific) using 96-well fast PCR Plates (Sarstedt) and Fast SYBR™ Green Master Mix (Applied Biosystems, ThermoFisher Scientific). The primers employed are listed in Supplementary Table 3.

**Plasmid transfection and cloning.** Cells were transfected with plasmids utilizing Lipofectamine 2000 (Invitrogen) in accordance with the manufacturer's recommendations. The MS2 loop was inserted into the scaRNA2 plasmid with the QuikChange II XL Site-Directed Mutagenesis Kit (Agilent). All plasmids and primers used for transfection, mutagenesis, cloning and PCR are listed in Supplementary Table 3. Unless otherwise specified, the pGEMT-easy plasmid was used routinely to obtain overexpression of scaRNA2. The pGEMT-easy scaRNA2 plasmid contains full-length scaRNA2 flanked on the upstream side with 250 bp containing its own promoter and downstream with 80 bp sequences (i.e., −250/+500) and this entire sequence was cloned into the vector utilizing EcoRI restriction sites. Since the pGEMT-easy vector lacks a mammalian promoter, expression of scaRNA2 is driven by its own promoter.

**siRNA/GapmeR transfections.** siRNAs (15 nM) and GapmeRs (25 nM) were transfected into cells using Lipofectamine RNAiMAX (Invitrogen) and HiPerfect (Qiagen), respectively, in accordance with the manufacturer's recommendations. The siRNAs and GapmeRs employed are listed in Supplementary Table 3.

**Single-molecule RNA FISH.** Ten specific DNA probes covering the entire 420-nucleotide (nt) sequence of full-length scaRNA2 were utilized. Each probe was 70 nt long and included a sequence antisense to scaRNA2 (30 nt) flanked with 5′- and 3′-linker sequences (á 20 nt). DNA oligomers specific for each linker sequence and containing two fluorescent labels (see Supplementary Table 3) were used for visualization.

Cells were first grown on coverslips and then fixed with 4% paraformaldehyde (PFA) (Santa Cruz Biotechnology) for 10 min. After washing in PBS and 70% ethanol, the coverslips were placed in RNA wash buffer (25% formamide (Ambion), 2x SSC (Ambion) in nuclease-free water) for 5 min and subsequently incubated overnight at 30 °C with 10 µM scaRNA2 probes (dissolved in

hybridization buffer containing 10% dextran sulfate (Sigma-Aldrich), 25% formamide, 2x SSC, 0.02% BSA, 1 mg/ml E.coli tRNA (Sigma-Aldrich), and 10 mM ribonucleoside vanadyl complex (New England Biolabs)). The following day these coverslips were incubated in RNA wash buffer for 1 h at 30 °C and thereafter with 2 µM fluorescent probes in hybridization buffer for 2–3 h at 30 °C. Finally, the coverslips were exposed to RNA wash buffer for 1 h at 30 °C and washed twice with SSC before mounting with VectaShield mounting medium containing DAPI (Vector Laboratories).

**Immunofluorescence microscopy.** For this purpose, cells were grown on sterilized coverslips and thereafter fixed with 4% PFA for 15 min at room temperature, followed by permeabilization with 0.1% Triton X-100 for 5 min at room temperature. The coverslips were then treated with blocking buffer (2% BSA, 5% glycerol, 0.2% Tween20, 0.1% NaN₃) for 30 min at room temperature and incubated with primary antibody for 1 h at room temperature or overnight at 4 °C, followed by 30 min incubation with Alexa fluor-conjugated secondary antibody in blocking buffer. All antibodies utilized are listed in Supplementary Table 4. Finally, the coverslips were incubated with 0.1 µg/ml DAPI for 30 min, then washed with PBS for 30 min and mounted with VectaShield (Vector Laboratories) or Mowiol 4-88 (CSH protocols) mounting media.

Micronuclei were defined as extranuclear bodies with a diameter less than a third of the corresponding nucleus and showing the same intensity of staining and lying in the same focal plane as that nucleus.

In experiments designed to visualize WRAP53β IR-induced foci and accumulation of MRN or NHEJ proteins at breaks created by FokI, this visualization was enhanced by incubating the cells prior to fixation with cytoskeleton (CSK) buffer (10 mM pipes (pH 7.0), 100 mM NaCl, 300 mM sucrose, 3 mM MgCl₂, 0.7% Triton X-100) for 5 min at room temperature.

Images were acquired with a LSM700 confocal microscope (Zeiss), mounted on Axio observer.Z1 (Zeiss) equipped with a Plan-Apochromat 63x/1.4 oil immersion lens, and then processed with the Zen 2012 Black software (Zeiss). Images were analyzed in ImageJ (Java 1.8.0_172 64-bit) and CellProfiler (version 3.1.8)[81].

**Laser micro-irradiation.** Localized DNA damage was created by micro-irradiation of U2OS cells (pre-sensitized with 10 µM 5-bromo-29-deoxyuridine for 24 h at 37 °C) using a Leica DMI 6000B microscope (Leica) equipped with a pulsed nitrogen laser (20 Hz, 364 nm; Micropoint Ablation Laser System) and a Leica ×40 HCX PL APO/1.25-0.75 oil-immersion objective. To inflict localized DNA damage along lines, the iteration was set to 1 and laser output to 12–20% (150 × 1 pixel). Prior to microscopy, the culture medium was replaced with a medium free of phenol red.

**The FokI system.** The U2OS FokI cells contain a stably integrated LacO array and continuously express the mCherry-LacI-FokI protein fused to a destabilization domain (DD) and a modified estradiol receptor (ER) (ER-mCherry-LacI-FokI-DD)[82]. This enables inducible nuclear expression of ER-mCherry-LacI-FokI-DD upon exposure of these cells for 4 h to 1 µM Shield-1 ligand (which stabilizes the DD-domain and thereby induces expression of ER-mCherry-LacI-FokI-DD) (Clontech) and 1 µM 4-hydroxytamoxifen (4-OHT; which causes translocation of this protein into the nucleus) (Sigma-Aldrich). At least 50 cells were examined in each experiment

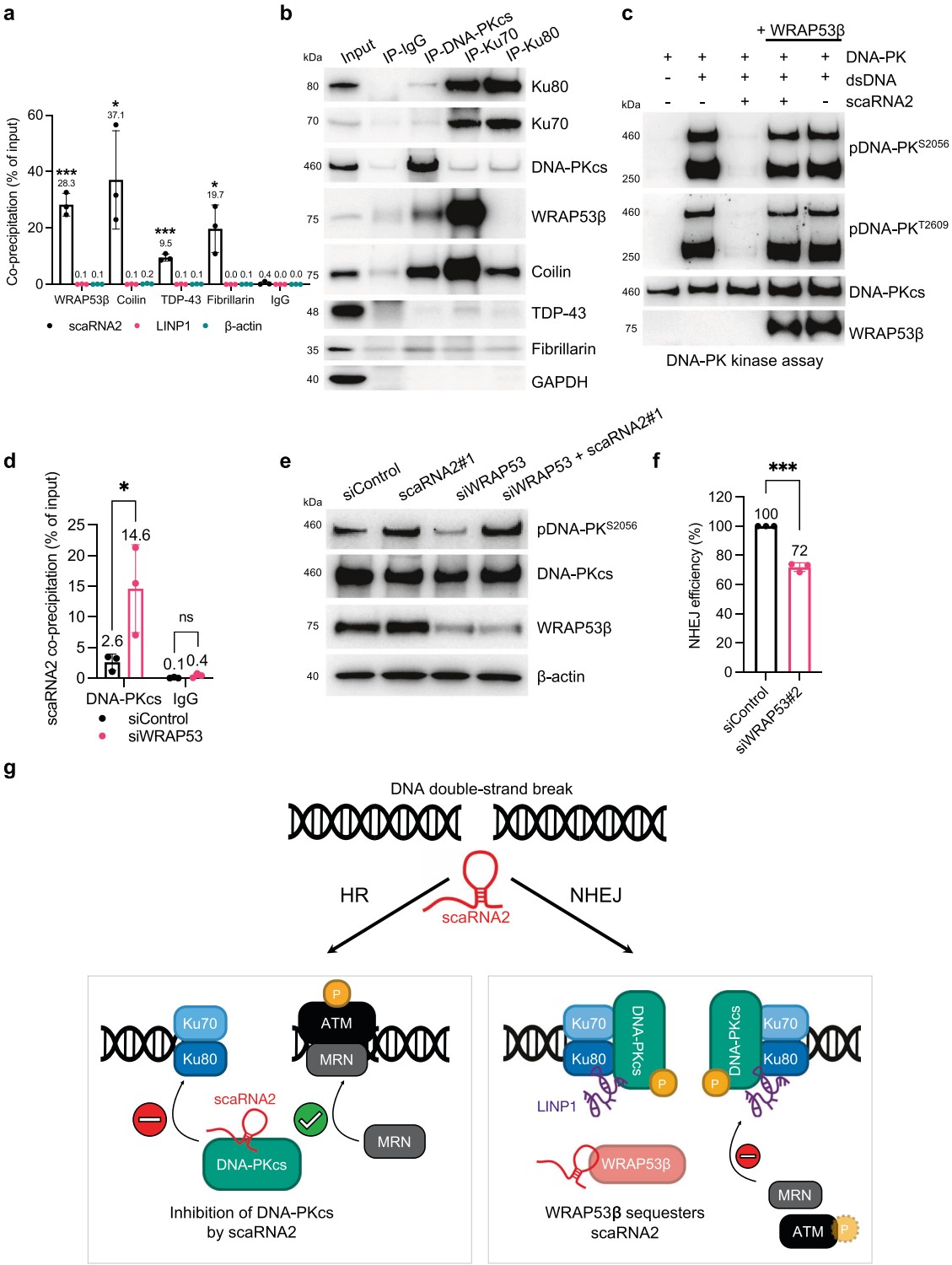

and the numbers on the graphs indicate the percentage of these cells where the repair factor co-localized with the mCherry-FokI dot.

**Ionizing radiation**. Cells were exposed to ionizing radiation (approximately 1.3 Gray (Gy)/min) from a CIX2 Xstrahl X-ray machine (Xstrahl) utilizing settings of 195 kV and 10 mA, a focus-to-specimen distance (FSD) of 40 cm, and a 3 mm aluminum filter. The exact doses and recovery times are documented in figure legends. At least 100 cells were examined in each experiment and the numbers on the graphs indicate the percentage of these cells whose nuclei contained at least 10 of the foci in question.

**CRISPR/Cas9 gene editing**. Both alleles of the *scaRNA2* gene were knocked out in MCF7 cells using the CRISPR- hSpCas9 system in the manner described in the Morrisey Laboratory Protocol for Generation of Large (>1 kb) Genomic Deletions Using CRISPRs (Dan Swarr and Dave Frank, Version 1, July 14, 2014). Short guide RNA (sgRNA) sequences targeting the 5' and 3' UTR sequence of this gene (Gene ID: 677766) were cloned into the pX459 vector (Addgene # 62988) and co-transfected using lipofectamine LTX (Invitrogen) into MCF7 cells stably expressing Cas9. Twenty-four hours after transfection, the cells were harvested and the knock-out efficiency assessed using primer pairs for genotyping. The cells were then counted and seeded at a density of 0.5 cells/well onto 96-well plates. Four weeks later, expression of scaRNA2 was analyzed by RT-qPCR, and a monoclonal cell line in which scaRNA2 had been knocked-out was used in the experiments.

**Fig. 7 Binding to and inhibition of DNA-PKcs by scaRNA2 is regulated by WRAP53β. a** CLIP of WRAP53β, coilin, TDP-43, fibrillarin or IgG from U2OS cells. The graphs show the amount of co-precipitated RNA as percentage of input (means ± SD $n = 3$) measured by qPCR, significant differences between the RNA of interest and β-actin are depicted, *$p < 0.05$, ***$p < 0.00$ as determined by unpaired two-tailed t-test. Representative blots depicting the immunoprecipitation of WRAP53β, coilin, TDP-43, fibrillarin or IgG are presented in Supplementary Fig. 10a. **b** IP of DNA-PKcs, Ku70, Ku80, or IgG from U2OS cells. The western blots depict the levels of co-precipitated proteins. The blots shown are representative of 3 independent experiments. **c** In vitro assay of DNA-PK kinase activity employing recombinant human DNA-PK (25–30 U), dsDNA (10 ng), scaRNA2 (100 ng) and/or WRAP53β protein obtained by IP from U2OS cells, followed by western blotting of the proteins indicated. The blots shown are representative of 3 independent experiments. **d** RIP of DNA-PKcs or IgG from U2OS cells transiently depleted of WRAP53β as indicated. The graph shows the amount of co-precipitated RNA as percentage of input (means ± SD $n = 3$) measured by qPCR, *$p < 0.05$, ns (not significant) as determined by unpaired two-tailed t-test. **e** U2OS cells were depleted of scaRNA2 and/or WRAP53β for 48 h, then irradiated (2 Gy) and 30 min later their proteins extracted for western blotting. The blots shown are representative of >3 independent experiments. **f** The efficiency of NHEJ following depletion of WRAP53β from U2OS EJ5-GFP NHEJ cells for 48 h. GFP expression was analyzed by flow cytometry. The values shown are means ± SD $n = 3$, ***$p < 0.001$ as determined by unpaired two-tailed t-test. **g** Our proposed model depicting the involvement of scaRNA2 in the cellular choice of pathway for the repair of DNA double-strand breaks. ScaRNA2 inhibits the association between DNA-PKcs and Ku70/80, thereby favoring recruitment of MRN and activation of ATM which facilitates repair by HR. WRAP53β can sequester scaRNA2, thereby preventing this RNA from binding to DNA-PKcs. The DNA-PKcs-Ku70/80-LINP1 complex can then form, leading to decreased ATM activation and reduced MRN recruitment, subsequently allowing NHEJ to proceed. In connection with DNA-PK regulation, scaRNA2 and LINP1 play opposing roles. Source data are provided as a Source Data file.

In the case of the U2OS cells, Cas9-expressing vectors were created with the GeneArt™ CRISPR Nuclease Vector with the orange fluorescent protein (OFP) Reporter Kit (Life Technologies). To obtain a genetic deletion of the *scaRNA2* gene, single sequences on both sites of the gene was chosen and these sgRNA target sequences then separately cloned into separate CRISPR Nuclease Vector. The two vectors (containing Cas9, OFP and either the 5′ or 3′ sgRNA sequence) were transfected into U2OS cells with lipofectamine LTX (Invitrogen) and cells were sorted 72 h later based on OFP expression. Following recovery for four days, single cells were isolated by serial dilution and cloned for genetic screening and characterization of expression. A monoclonal cell line in which scaRNA2 had been knocked-out and which lacked residual expression of Cas9 was used in the experiments. The sgRNAs and PCR primers employed are listed in Supplementary Table 3.

**Western blotting**. Cells were lysed in NP40 lysis buffer (100 mM Tris-HCl, pH 8, 150 mM NaCl, 1% NP-40, 1% protease inhibitor cocktail) for 30 min on ice. DNA was sheared by sonication for 4 cycles (30 sec ON/30 sec OFF, medium intensity) in a Bioruptor (Diagenode) followed by centrifugation at $16000 \times g$ for 15 min at 4 °C and the protein concentration of the supernatant thus obtained determined using the Bradford assay (Bio-Rad). The proteins in these extracts were subsequently resolved on NuPAGE® precast gels (Life Technologies) and transferred onto nitrocellulose membranes, which were incubated with primary antibodies (Supplementary Table 4) followed by horseradish peroxidase-conjugated secondary antibodies (Cell Signaling). The antibodies directed towards the total protein were diluted in 5% milk containing PBS-Tween20, whereas antibodies towards the phosphorylated forms were diluted in 5% BSA (Sigma-Aldrich) containing PBS-Tween20. The western blots were developed using the SuperSignal West Femto maximum sensitivity substrate (Thermo Fisher Scientific) in accordance with the manufacturer's instructions.

**Immunoprecipitation**. Native IP: Lysates were prepared as described for western blotting. 100–600 µg total protein was incubated with 5–10 µl Dynabeads™ Protein G (Invitrogen) and 1 µg antibody in a total of 600 µl in NP40 buffer overnight with rotation at 4 °C. Thereafter, the beads were washed twice with NP40 and twice with RIPA buffer (50 mM Tris-HCl, pH 7.4, 150 mM NaCl, 2 mM EDTA, 1% NP-40, 0.5% sodium deoxycholate, 0.1% SDS) for 15 min each time and prepared for blotting. Where indicated, RNase A was included during the immunoprecipitation at 0.2 mg/ml.

RNA immunoprecipitation (RIP): RNA was extracted from the beads with TRIzol® (Life Technologies, ThermoFisher Scientific), 1-bromo-3-chloropropane (Sigma-Aldrich) and the RNeasy Mini Kit (Qiagen), following the manufacturer's instructions.

CLIP: U2OS cells were cross-linked by exposure to UV light (254 nm, 150 mJ/cm² in a CL-1000 Ultraviolet Crosslinker) prior to lysis as described for western blotting. 100 µg total protein was incubated together with 5 µl Dynabeads™ Protein G (Invitrogen) and 1 µg antibody in a total of 600 µl in NP40 buffer overnight with rotation at 4 °C. Thereafter, the beads were washed twice with cold high-salt buffer (1 M NaCl, 0.05 M Tris-HCl, 1 mM EDTA, 1% NP40, 0.10% SDS, 0.50% sodium deoxycholate) and twice with cold CLIP wash buffer (10 mM MgCl₂, 20 mM Tris-HCl, 0.20% Tween 20) for 10 min each time, before being washed once again with cold NP40 buffer. For isolation of RNA, the beads were first incubated with 4 U Proteinase K (Molecular Biology Grade, New England Biolabs; diluted in Proteinase K buffer (0.01 M Tris-HCl, 0.1 M NaCl, 1 mM EDTA, 0.20% SDS) to give a total volume of 100 µl) for 60 min at 50 °C and shaking at 1100 rpm, followed by addition of TRIzol® (Life Technologies, ThermoFisher Scientific) and trichloromethane (Sigma-

Aldrich), with subsequent extraction of RNA utilizing the RNeasy Mini Kit (Qiagen), in accordance with the manufacturer's instructions.

IP in vitro: Recombinant DNA-PK (120U/reaction, Promega) and recombinant Ku70/80 (0.25 ug/reaction, Sino) were diluted in 50 ul 1x EMSA buffer (10 nM HEPES pH7–5, 20 mM KCl, 1 mM MgCl₂, and 1 mM DTT) in the absence or presence of 100 ng linearized plasmid or 100 ng scaRNA2 transcribed in vitro. These mixtures were incubated for 30 min at 30 °C to allow protein/RNA/DNA binding, followed thereafter by addition of 0.2 ug 18-2 antibody, incubation for an additional 30 mins and, finally, addition of 2.5 ul dynabeads and rotation of the mixture for 1 h at room temperature. The beads were washed 3 times with NP40 and prepared for western blotting.

**HR and NHEJ reporter assays**. U2OS DR-GFP HR[50] and U2OS EJ5-GFP NHEJ[83] reporter cells were first transfected with the siRNAs or GapmeRs indicated and 8 h later with 1 µg of an I-SceI vector alone or in the case of rescue or overexpression, in combination with a scaRNA2 plasmid. Forty-eight hours after the initial transfection, the cells were harvested in PBS and fixed in 4% PFA for 15 min. The GFP signal associated with repair events was assessed by flow cytometry on a BD LSR II (BD Biosciences) and subsequently analyzed using BD FACSDiva software v8.0.2 (BD Biosciences), with quantification carried out utilizing FlowJo v9 (BD Biosciences).

**Cell cycle analysis and synchronization**. U2OS DR-GFP cells were transfected with siRNA or GapmeRs for 48 h, fixed with cold 70% ethanol overnight at −20 °C and thereafter stained with propidium iodide (0.05 mg/ml (Sigma-Aldrich) plus 0.25 mg/ml RNase A in PBS) for 30 min at 37 °C with protection from light. The samples were then examined by flow cytometry on a BD LSR II (BD Biosciences), followed by analysis with BD FACSDiva software v8.0.2 (BD Biosciences) and quantification using FlowJo v9 (BD Biosciences).

To achieve synchronization, cells were incubated with 2 mM thymidine (Sigma-Aldrich) for 18 h, incubated without exposure for 8 h, and then subjected to the same thymidine treatment a second time (2 mM for 18 h). Cells in G1 phase were collected immediately after this second thymidine treatment (0 h) and cells in the S and G2-phases were collected 4 and 8 h thereafter, respectively. The phases of the cell cycle were analyzed by flow cytometry and the levels of specific species of RNA by qPCR.

**Luciferase splicing assay**. The Bright Glo luciferase assay was performed on HeLa Luc and HeLa Luc-I cells as described previously[84,85]. As a positive control, these cells were treated with 100 nM of the splicing inhibitor Pladienolide B (Tocris Biosciences) for 16 h.

**DNA-PK kinase assay in vitro**. The assay was performed in 1x kinase buffer (Thermo PV3189) containing 200 µM ATM, 80 µg/ml BSA and 25–30 U recombinant DNA-PK holoenzyme (Promega V581A) in a final volume of 30 µl. Where indicated, 1–100 ng of in vitro transcribed scaRNA2, 10 ng linearized DNA plasmid, WRAP53β protein and/or recombinant Ku70 and Ku80 (SinoBiological CT018-H07B) was added. All reactions were allowed to proceed for 60 min at 30 °C and the results subsequently assayed by western blotting.

In vitro transcription: Following linearization and digestion with proteinase K (New England Biolabs), scaRNA2, 5, and 7 were transcribed in vitro from plasmids encoding these RNAs using the MEGAscript T7 transcription kit (Thermo Fisher Scientific) in accordance with the manufacturer's instructions. A LINP1 fragment with a T7 promoter was generated by PCR and subsequently transcribed in vitro as described above. Linear, double-stranded DNA was prepared from the Flag-

pCMV-Tag2B-empty plasmid and then digested with BamHI and HindIII and purified using the QIAquick PCR purification kit (Qiagen). WRAP53β protein was immunoprecipitated from U2OS cells (untreated or depleted of Cajal body factors for 48 h) and the beads were then washed extensively with NP40 buffer containing 400 mM NaCl, 50 mM Tris, and 1% NP40 for 30–60 min at 4 °C to remove contaminating proteins and subsequently treated with 300 µg/ml RNase A for 20 mins at 37 °C to remove contaminating RNA. Thereafter, the beads with bound WRAP53β were dissolved in 20 µl EMSA buffer (10 mM HEPES, pH 7.5, 20 mM KCl, 1 mM MgCl$_2$, and 1 mM DTT) and assayed immediately for DNA-PK kinase.

**Statistics and reproducibility**. Unless otherwise indicated, all values presented are the means ± standard deviations and $n = 3$ refer to three biologically independent experiments. The Prism GraphPad software v9.2.0 was used for all analyses and the different groups compared for statistically significant differences employing unpaired two-tailed Student's t-test or one-way ANOVA followed by two-sided Dunnett's or Šidák's multiple comparisons test. $p < 0.05$ was considered to be significant and the $p$-values are provided as follows: \*$p < 0.05$, \*\*$p < 0.01$, \*\*\*$p < 0.001$, ns (not significant).

**Reporting summary**. Further information on research design is available in the Nature Research Reporting Summary linked to this article.

## Data availability

All other data supporting the findings of this study are available from the corresponding author on reasonable request. Source data are provided with this paper.

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

## Acknowledgements

We wish to thank Dr Tamas Kiss (University of Toulouse) for the pGEMT-easy scaRNA2 plasmid, Dr Hebert for the plasmids expressing scaRNA2 deletion variants, Dr Roger Greenberg for the U2OS-FokI cell line, Dr Helleday for the HR reporter cells, Dr Stark for the NHEJ reporter cells and Dr Galina Selivanova for the MCF7 cells expressing Cas9. We would also like to thank Dr Magda Bienko (Karolinska Institutet) for sharing her protocol for single-molecule RNA FISH with us and Karen Akopyan, Arne Lindqvist and Florian Salomons for helpful input on image analysis and laser microscopy. This work was supported by grants from the Swedish Cancer Society (MF), the Swedish Research Foundation (MF), the Strategic Research Programme in Cancer (MF), the Center for Innovative Medicine (MF), Radiumhemmet Foundation (MF), Wenner Gren Foundation (MF) and the funds of Karolinska Institutet (MF). These organizations played no role in the study design, data collection or analysis, our decision to publish or preparation of the manuscript.

## Author contributions

All of the authors S.B., E.O.B., C.C., D.H., D.P., S.S., S.D., C.P., L.S., O.M., J.O.R., and M.F. were involved in performing the experiments described and provided intellectual input. M.F. and C.C. initiated the study, L.S. generated the CRISPR cell lines, J.O.R. performed the splicing assay, S.B., E.O.B., C.C., and M.F. designed experiments, analyzed data, and wrote the paper.

## Funding

## Competing interests

The authors declare no competing interests.
