## [Peer Review File · Nature Communications]

Title: Small Cajal body-associated RNA 2 (scaRNA2) regulates DNA repair pathway choice by inhibiting DNA-PKEditorial Note: Parts of this Peer Review File have been redacted as indicated to maintain the confidentiality of unpublished data.

REVIEWER COMMENTS

Reviewer #1 (Remarks to the Author):

Review of Small Cajal body-associated RNA 2 (scaRNA2) regulates the choice of DNA repair pathway by inhibiting DNA-dependent protein kinase (DNA-PK) by Bergstrand and colleagues.

In this work, the authors have provided compelling evidence of a novel regulatory pathway for DNA-dependent protein kinase that is mediated by scaRNA2. The paper is very well-written, the data and methods are comprehensive, and the conclusions are, for the most part, supported by the presented data. In addition to being a pleasure to read, the manuscript is appropriate for the broad readership of Nature Communications.

With that being said, there are a few major issues with the manuscript that I believe need to be clarified before publication. In addition, I have a few minor concerns.

Major issues:

1) All the data in the manuscript are incorporated into the model shown in Fig. 7E. In this model, and elsewhere in the manuscript, the importance of the WRAP53 protein (TCAB1/WDR79) in directly interacting with scaRNA2 is emphasized. The justification for exploring the WRAP53 protein in regulating the putative role of scaRNA2 in DNA repair comes from cited previously published work (refs 22, 25, 37 and 38) supposedly showing an interaction between WRAP53 protein and scaRNA2 (lines 97 and 98). However, these cited publications (refs 22, 25, 37 and 38) DO NOT show that WRAP53 protein directly interacts strongly with scaRNA2. In fact, two of these publications make it clear that WRAP53 protein binds to the CAB (Cajal body-localization) motif present in box H/ACA scaRNAs (including the CAB motif found in hTERC) and does not strongly interact with box C/D scaRNAs, which lack a CAB motif. ScaRNA2 is one such box C/D scaRNA that lacks a CAB motif.

Here is what each of these references show about WRAP53 protein binding to box C/D scaRNAs that lack a CAB motif:

Reference 22 (Tycowski, 2009): The authors state “Unlike H/ACA and mixed domain C/D-H/ACA scaRNAs, human C/D scaRNAs were only marginally coprecipitated with the ectopically expressed Myc-hWDR79 fusion (data not shown).”

The authors go on to state that they were able to show C/D scaRNA interaction with WRAP53 protein by

utilizing a more stringent lysis condition. One interpretation of this result is that the WRAP53 protein does not directly interact with box C/D scaRNAs but instead this association is mediated by another protein factor that is liberated in the more stringent lysis conditions.

In lines 107 and 108 of the manuscript under review, Bergstrand et al. state that scaRNA2 is, for unknown reasons, more than 90% associated with chromatin (Fig. 1A). This finding further emphasizes that scaRNA2 likely associates with other proteins that are associated with chromatin and these chromatin proteins can be part of the WRAP53 protein complex.

Reference 25 (Bergstrand, 2020): scaRNA2 is co-precipitated with GFP-WRAP53 protein (but not with point mutations of WRAP53 which cause Hoyerdal-Hreidarsson syndrome). GFP-WRAP53, but not point mutations of WRAP53, was also found to interact with proteins enriched in the Cajal body such as coilin and SMN. A direct interaction between WRAP53 protein and scaRNA2 is not shown.

Reference 37 (Venteicher, 2009): This paper does not evaluate direct binding of WRAP53 protein to scaRNAs, but utilizes IPs to determine if scaRNAs co-precipitate with the WRAP53 protein complex. Box C/D scaRNAs were not evaluated in this paper. Rather, the authors examined if WRAP53 protein associates with mixed domain scaRNAs (scaRNA10 and scaRNA5) in addition to box H/ACA scaRNAs (scaRNA8 and scaRNA13), all of which have a CAB motif.

Reference 38 (Izumikawa, 2019): The authors state that “WDR79 (WRAP53 protein) binds C/D scaRNAs several-fold less efficiently than do H/ACA scaRNAs (32,33,35), such that involvement of other unknown factor(s) in the CB localization of C/D scaRNAs has been proposed (32,33,36). In this study, we identified TDP-43 as a major player that regulates the CB localization of a subset of C/D scaRNAs independently of WDR79.”

The authors go on to state that TDP-43 is the major factor that targets scaRNA2 to Cajal bodies and WRAP53 protein interaction with scaRNA2 is dependent on TDP-43.

2) Collectively, these and other publications strongly suggest that WRAP53 protein interaction with scaRNA2 is likely mediated by other components enriched in the Cajal body, such as TDP-43 and coilin. Hence these other proteins (TDP-43, coilin and possibly others) are probably part of the WRAP53 protein complex and facilitate a subset of box C/D scaRNA interaction, such as scaRNA2. This should be at least discussed if not tested experimentally.

3) Until the authors formally exclude the role of TDP-43, coilin, and other CB-enriched proteins in mediating the association of scaRNA2 in the WRAP53 protein complex, the authors need to revise the manuscript text and model shown in Figure 7E. Ideally, the authors could clarify that the WRAP53 complex contains other proteins which may directly interact with box C/D scaRNAs that lack a CAB motif or GU/UG wobble stem. (The GU/UG wobble stem is another motif (Marnef, 2014) shown to interact with the WRAP53 protein). ScaRNA2 is one example of a scaRNA lacking a CAB motif and a GU/UG wobble stem and hence there is a strong possibility that other proteins in the WRAP53 complex mediate

this interaction.

4) Along these lines, a quick PubMed search shows that coilin has been shown to interact with Ku proteins and inhibit non-homologous DNA end joining (Velma, 2011) as well as be recruited to UVA-induced DNA lesions (Bartova, 2014). This information is very relevant to the manuscript and should be discussed. Since coilin is part of the WRAP53 complex, have the authors examined if coilin contributes to any of the proposed WRAP53 protein activities?

Minor issues:

1) The authors should briefly introduce box H/ACA, box C/D and mixed domain scaRNAs as well as the CAB motif, the GU/UG wobble stem and scaRNP formation and function. ScaRNA association with TDP-43 and coilin should also be discussed. It is highly likely that the scaRNA2 activity discussed in the manuscript utilizes scaRNA2 packaged into a RNP or complexed with other proteins. The authors report that 2-3% of scaRNA2 co-precipitates with DNA-PKcs (line 277). Do known box C/D scaRNP proteins such as fibrillarin also co-precipitate with DNA-PKcs?

2) Line 115 states that scaRNA2 overexpression increases CB numbers (Fig. 1B), but no quantification is shown. Also, a negative control RNA overexpression is not shown to monitor for increased CB numbers as a consequence of increased transcription. In addition, no apparent increase in CB number is seen in Fig. S1F which also shows overexpressed scaRNA2. These points need to be addressed in order to support the statement in line 115.

3) The scaRNA2 expression plasmid was obtained from the Kiss lab per the information in one of the supplemental tables. However, this plasmid has the pGEMT-easy backbone which does not have a mammalian promoter. Please include more details about the scaRNA2 expression vector. This information is relevant given that scaRNA2, unlike typical pol II-derived mRNAs, likely has both promoter and terminator elements that influence its expression and processing into the nucleolus-enriched mgU2-61 fragment (Tycowski et al. 2004).

4) Figure S1: The schematic of scaRNA2 is incorrectly annotated. According to Tycowski et al. 2004, the mgU2-61 domain consists of a C, D', C', and D box that organize into a stem-loop structure. Additionally, Izumikawa et al. (2019) found that a UG-rich motif upstream of mgU2-25 is necessary for TDP-43 binding and trafficking to the CB. The authors should edit the schematic to more accurately depict vital elements of the structure of scaRNA2.

5) What happens to CBs and CB numbers when scaRNA2 is reduced?

6) In Fig. 4, the arrangement of panel E being up by panel A is not optimal.

7) In Fig. 5, have the authors tested a negative control RNA? Would the inclusion of any RNA reduce DNA-PK autophosphorylation? It is possible that the in vitro system might be more sensitive to any RNA

whereas in cells only RNA complexes that interact with DNA-PK (such as a scaRNP2 complex) reduce autophosphorylation.

8) In Fig. 7B, the WRAP53 protein was obtained by IP from U2OS cells. Consequently, proteins in the WRAP53 complex (such as TDP-43 or coilin) may be responsible for sequestering away the scaRNA2 resulting in the increase of DNA-PK autophosphorylation. The authors can easily test for the potential contribution of CB-enriched proteins in this assay by IPing WRAP53 from cells treated with siRNAs to TDP-43 or coilin or both. Alternatively, TDP-43 or coilin IP complexes can be used in this assay. For a more direct assessment of the regulation of WRAP53 in this assay, WRAP53 would need to be bacterially purified to ensure that other CB-enriched proteins are not present in the complex used for the assay.

9) Fig. S4 D and E, lines 272-281. Based on the input signals, there is not much IP of DNA-PKcs with the 18-2 antibody. Hence it is hard to say that the epitope to which this antibody binds mediates interaction with scaRN2. A first step could be to RNase treat the lysate before 18-2 antibody IP to see if this increases DNA-PKcs recovery.

Reviewer #2 (Remarks to the Author):

Bergstrand, O'Brien et al. describes the involvement of scaRNA2 in the regulation of DNA double-strand break repair pathway choice. They propose that scaRNA2 regulates DNA repair pathway choice by limiting DNA-PK activation through direct binding of scaRNA2 to the DNA-PKcs. This is proposed to stimulate Homologous Recombination, through blocking DNA-PK assembly, limiting NHEJ, promoting ATM activation and eviction of the NHEJ proteins. In the last part of the manuscript, they also involve WRAP53beta as a regulator of scaRNA2 activity: WRAP53beta is proposed to sequester scaRNA2 away from DNA-PKcs, thereby promoting efficient NHEJ. However, while the model proposed is really appealing, it is currently not based on sufficient experimental evidences. Multiple important information are missing and key aspects of the model (e.g. the direct binding of scaRNA2 to DNA-PKcs) need to be consolidated by additional experiments. Considering the large amount of experiments required to support the proposed role of scaRNA2 in DNA repair pathway choice, we feel the manuscript should be rejected.

Main comments:

- In the abstract, the authors claim that "cells lacking scaRNA2 show a pronounced reduction of the level of the MRN complex". However, the data in Extended data Fig.S2G (transient knock-down of scaRNA2) do not support that claim. In fact the title of Extended Fig. S2 is "Loss of scaRNA2 does not alter the cell cycle or expression of repair proteins". Consequently the authors should remove that sentence from the abstract. In addition, they should blot for MRE11, RAD50 and NBS1 in the stable scaRNA2 conditions to complement the right panel of Extended Fig. S2G.

- In the abstract, the authors claim that “Cells lacking scaRNA2 exhibit a pronounced reduction of (...) activation of ATM at DNA ends”. However the data do not support that claim: The effects on ATM activation as monitored after IR with PhATM and gammaH2AX are very mild: a reduction of 25% in foci intensity. The abstract need to be rephrased to convey this moderate decrease of ATM activation. The effect observed using the LacI-FokI system could be explained by reduced cutting activity of the FokI enzyme, since the control of DSB induction used here for this system is also gammaH2AX, which is itself reduced by scaRNA2 down-regulation, suggesting reduced cutting.

- What controls scaRNA2 recruitment at sites of DNA damage? The authors should deplete DNA-PKcs to test how this impacts on scaRNA2 recruitment at laser lines (since based on their model, scaRNA2 associates with DNA-PKcs), and test different inhibitors (ATM, ATR, DNA-PKcs and PARP inhibitors) to determine the dependencies for the observed recruitment.

- The pictures that are shown on the side of multiple graphs do not fit with data from the quantification shown on the graphs. The authors have to show pictures with 2-3 cells instead of a single one, which is not enough to show the heterogeneous response observed. For example, showing a single cell cannot convey the decrease from 42% positive cells to 25% positive observed for RPA2 on the Fig.2A panel. Accordingly, pictures have to be replaced in Fig.2A, Fig.2B, Fig.2C, Fig.2D, Fig.3A, Fig.3B and Fig. 4B, to fit to the respective quantifications.

- In the abstract, the authors claim that “blockade of either the assembly or catalytic activity of DNA-PK in cells lacking scaRNA2 restore HR”. However this is not what has been done in the manuscript: DNA-PK inhibition restored RAD51 loading, a step in the HR process, but not a readout of HR efficiency. The abstract should be modified accordingly. In addition, a rescue by the blockade of DNA-PK assembly was not performed (The experiment with LINP1 is indirect). This should be removed from the abstract or experiments aimed at blocking DNA-PK assembly (such as siRNA against DNA-PKcs) should be performed.

- The blots in figure S3 are incomplete since in each case, the other subunits of the DNA-PK complex have to be blotted (most probably, the whole complex is present in each IP because of contaminant broken DNA in the extracts). Unless this, nothing can be concluded on the respective affinity of scaRNA2 for DNA-PKcs or Ku. Explain why a co-IP between Ku70 and scaRNA2 is observed but not with Ku80 ? (Figure 4A) is the antibody interfering with scaRNA2 binding ? Is the epitope of this antibody mapped ? This could provide insights into the mode of interaction between Ku and scaRNA2, and suggest that scaRNA binds to Ku80 and not only to DNA-PKcs. When an inhibition of scaRNA2 binding was observed with the 18-2 anti-DNA-PKcs antibody, this is how it was interpreted. One can propose that the antibody against Ku80 blocks the interaction with DNA-PKcs, which would explain why scaRNA2 was not found in the Ku80 IP, but then this should be tested properly by comparing the fraction of DNA-PKcs immunoprecipitated by the Ku70 and Ku80 antibodies as compared to control IP.

- More mechanistic insights should be provided regarding the mode of interaction of scaRNA2 and DNA-PK. Is scaRNA2 binding to Ku or DNA-PKcs. This could be tested by performing RIP to check for scaRNA2

association to Ku70 in DNA-PKcs depleted cells. If the observed scaRNA2-Ku co-IP depends of DNA-PKcs this should be reduced in DNA-PKcs depleted cells.

- Mapping on DNA-PKcs (and Ku) the interaction domains of scaRNA2 should be performed. UV-mediated crosslink coupled with MS/MS could be used for this experiment for example.

- In the in vitro experiments using in vitro synthesized scaRNA2, a control structured RNA should be used (for example TERC/scaRNA19) to confirm a specific effect, since Ku binds secondary DNA/RNA structures such as hairpins, stem loops, etc. The simplest interpretation of data in Figures 5A, B is that Ku is titrated out by binding to scaRNA2, possibly non-specifically.

- Which part of scaRNA2 is important for this activity ? Various deletions should be tested (based on the predicted or known secondary structures) by complementing the scaRNA2 KO cells and monitoring its recruitment to sites of DNA damage (laser+smFISH or scaRNA2-MS2 as used in extended Fig.S2A).

- To support the model, the authors have to show that scaRNA2 depletion and overexpression affects cells survival to DNA damaging agents. From the proposed model of the implication of scaRNA2 into the DNA damage response, scaRNA2 overexpression (inhibiting NHEJ) should sensitize to IR, but not to drugs inducing DSB mainly repaired by HR such as DSBs induced by camptothecin (CPT). In contrast, scaRNA2 depletion (inhibiting HR) should sensitize to both IR and CPT. This should be tested.

- In Fig. 5D the pattern obtained with the Phospho S2056-DNA-PKcs antibody does not seem specific (pan nuclear phosphorylation, increased in the nucleoli ?), knowing the reported frequent cross-reactivity of common Phospho S2056-DNA-PKcs antibodies (e.g. with Ph 53BP1). A control with NU7441 should be added to confirm that the staining is specific. If not, remove the figure 5D.

- The authors should show the level of spontaneous DNA damage (number of gammaH2AX foci) in cells depleted (or KO) for scaRNA2 or overexpressing scaRNA2. Defective HR usually translates into increased basal DNA damage. In addition to this, scoring micronuclei could also support an increased genomic instability induced by modulation of scaRNA2 levels.

- The authors should show that the decrease in NHEJ activity induced by WRAP53 depletion can be rescued by depletion of scaRNA2 to connect scaRNA2 to the observed inhibition.

- The model (Fig. 7E) is confusing since it suggests that MRN binding and Ku-DNA-PKcs binding are mutually exclusive while they are not since they have a different recruitment mechanism to DSBs (MRN binding on the side of the DSB, while Ku binding at the DNA ends through accommodating DNA end in its cavity). The model should be clarified to stick to the findings: scaRNA2 inhibits DNA-PK activity and thereby promotes HR. scaRNA2 inhibits the association between Ku and DNA-PKcs (coIP Fig.6A), thereby promoting Ku eviction and HR, but it also contributes to the recruitment of CtIP and MRN (Fig.3A).

Minor comments:

- Extended Figure 1A is not clear. The same colors are used to present different features (green bar for both MS2 and smFISH probes). The MS2 is inserted into the RNA, therefore a symbol representing this insertion would be used rather than a bar which suggests that the MS2 box is already present in the RNA.
- In the Results section, indicate whether MCF7 and U2OS cells KO for scaRNA2 correspond to individual clones or to a population.
- From the Mat & Med, it seems that the WT and scaRNA2 KO U2OS and MCF7 constitutively express Cas9. This should be indicated in the figure legend.
- Line 166-167. The accumulation of MRN and activation of ATM are not “defective” but “reduced”. Modify accordingly. In fact the reduction is quite mild.
- Figure 3 legend: again “loss of scaRNA2” does not “impairs” recruitment of MRN and ATM activation but “reduces” it. Modify accordingly.
- What are scaRNA2 secondary structures ? (predicted or known) Considering the affinity of Ku for hairpins, this should be discussed and if relevant presented in a supplementary figure.
- Is scaRNA2 transcribed by RNA polymerase II ? This should be stated when describing it the first time. Is it spliced ? Again this should be stated. scaRNA2 should be in bold in Table S1, since it is the focus of the study.
- The use of the “in vivo” expression should be avoided since it suggests experiments in animals. For clarity, replace by “in cells”.
- Extended Fig S2G, right panel: scaRNA2 KO cells show an increase in 53BP1 and BRCA1 levels. This should be discussed in the text.
- Pladienolide B is used in Extended Fig. S2I. In the Figure legend, indicate what are the target/effects of Pladienolide B.
- Indicate in the Figure 5C and Extended data Fig. S5 legends what the star on the right corresponds to (unspecific band of which antibody ?)

Reviewer #3 (Remarks to the Author):

The manuscript by Bergstrand et al., describes how lncRNA Scar2 regulates DNA repair pathway choice by inhibiting NHEJ via DNA-PK inhibition and consequently promoting HR. This is further regulated by Scar2 binding competition between DNA-PK and WRAP53B.

This story describes novel and interesting mechanism providing another evidence for RNA dependent DDR playing role not only in cis but also in trans. However, some experiments are missing key controls.

Here are my comments:

Figure 1:

A: adding other RNA controls would help, for example to compare to NEAT1 and another mRNA as cytoplasmic marker

B: This experiment is missing control with scar2 GAPmer; could you add laser potency and iterations to material and methods?

C: non-specific smFISH probes should be used as negative control, in the same laser setting; 24% also seems quite low, does this mean that in 76% of damaged cells scar2 doesn't localise to the breaks? What is an explanation for this? Perhaps just transient localisation? In this experiment, scar2 is overexpressed from plasmid, could authors repeat this with endogenous levels of scar2?

D: To be honest I am not convinced by these images: to me the green and red dots do not really overlap for scar2 (as they do for gH2AX) Did the authors test that adding MS2 hairpin is not affecting scar2 function/localisation; could they move it to another region on scar2, where it's not close to C and D box structures? What are the working concentration for Shield and 4OH?

E: the image for scar2/3 doesn't really correspond to quantifications provided, it looks like there is more gH2AX foci at 24 hours when compared to 1 hour time point; are they swapped??

What are the levels of other scaRNA when they use GAPmers for scar2? Can authors provide some information on off target effect?

Could they show levels of scar2 after 3day post GAPmer transfection?

F,G: it would be interesting to know what is the phenotype for scar2 KO cells. What passage they are? In principle there could be some mutations accumulating in time, due to upregulated NHEJ and low HR. It would be also interesting to see if this effect on NHEJ sensitise them to PARPi. How in these KO cells gH2AX looks like at 0 (no irradiation)?

Figure 2:

A and B: could the authors use scatter plots, comparing mean but also showing cell distribution? Also the authors should supplement these data scar2 overexpression/rescue.

It would be great to show that lack of scar2 also affect endogenous HR sites. The authors could show less of resection in sequence specific system.

I am also surprised that the authors didn't see more pronounced effects on 53BP1 as one would expect based on recent study <https://www.nature.com/articles/s41467-019-12836-9>

D: could the authors repeat this rescue with scar2 lacking MS2 overexpression.

F: the authors should add scar2-MS2 here for comparison

Figure 3

C and D seems to be somewhat contradictory to Figure 1 E/F/G. Can the authors explain why in one figure they see more gH2AX foci in cells lacking scar2, but not in another??

It would good to add pATM and gH2AX to western blots.

B: could they test whether pATM recruitment is completely impaired or just delayed? More time after damage should be added.

These data could be further supported by ChIP at DSBs in AsiSI for example.

Figure 4

A: nuclear non-specific protein control is missing

B: 53BP1 should added

C: the authors need to add scar2 rescue/overexpression control and KD of known NHEJ factor as a negative control

D: scar2 KD should be added for comparison

Figure 5

A: non-specific RNA of the same length as scar2, ideally another scar should be added as a negative control

C: here the authors use 6GY, whilst in Fig 1 they used 2 GY for irradiation. Is there any explanation for it? It would be good to be consistent.

D: Scar2 overexpression/rescue should be added

E: they should add longer time point here, 1hour or longer

What the NS t-test corresponds to? Surely not to no IR and 5min. Is it shifted?

Figure S3

A and B Nuclear protein controls are missing

Figure 6

A, B and C: nuclear protein control is missing

What are the levels of LINP on chromatin after Scar2 depletion?

Furthermore, the authors show that scar2 prevents DNA-PK autophosphorylation and complex assembly. They also show that there is a competition between scar2 and LINP1 lncRNA, where scar2 is a negative regulator, whilst LINP2 is a positive one. It would be interesting to understand what determines the winner of this competition. Is expression of these lncRNAs cell cycle regulated, are they antagonistically expressed? If not, in which situation DNA-PK would preferentially bind to either scar2 or LINP1? Could author extent their studies to provide more clarifications for this?

Figure 7

S4D and E: these data should be supported with in vitro studies

B: could add LINP here?

Similarly, the authors show that WRAP53B is in competition with DNA-PK for scar2 binding. Again, how is this regulated? Any changes depending on cell cycle phase or perhaps in response to DNA damage? It would be great to understand this molecular mechanism a bit more.

Does WRAP35B bind to LINP1?

Does WRAP35B bind to DNA-PK?

Point-by-point response to the reviewers' comments

Dear Reviewers:

We appreciate the time and effort you have all spent reviewing our manuscript. Below, please find our detailed responses to your concerns.

Response to Reviewer 1

In this work, the authors have provided compelling evidence of a novel regulatory pathway for DNA-dependent protein kinase that is mediated by scaRNA2. The paper is very well-written, the data and methods are comprehensive, and the conclusions are, for the most part, supported by the presented data. In addition to being a pleasure to read, the manuscript is appropriate for the broad readership of Nature Communications. With that being said, there are a few major issues with the manuscript that I believe need to be clarified before publication. In addition, I have a few minor concerns.

Response: We thank the reviewer for this positive feedback.

Major issues:

1) All the data in the manuscript are incorporated into the model shown in Fig. 7E. In this model, and elsewhere in the manuscript, the importance of the WRAP53 protein (TCAB1/WDR79) in directly interacting with scaRNA2 is emphasized. The justification for exploring the WRAP53 protein in regulating the putative role of scaRNA2 in DNA repair comes from cited previously published work (refs 22, 25, 37 and 38) supposedly showing an interaction between WRAP53 protein and scaRNA2 (lines 97 and 98). However, these cited publications (refs 22, 25, 37 and 38) DO NOT show that WRAP53 protein directly interacts strongly with scaRNA2. In fact, two of these publications make it clear that WRAP53 protein binds to the CAB (Cajal body-localization) motif present in box H/ACA scaRNAs (including the CAB motif found in hTERC) and does not strongly interact with box C/D scaRNAs, which lack a CAB motif. ScaRNA2 is one such box C/D scaRNA that lacks a CAB motif.

Here is what each of these references show about WRAP53 protein binding to box C/D scaRNAs that lack a CAB motif:

Reference 22 (Tycowski, 2009): The authors state “Unlike H/ACA and mixed domain C/D-H/ACA scaRNAs, human C/D scaRNAs were only marginally coprecipitated with the ectopically expressed Myc-hWDR79 fusion (data not shown).”

The authors go on to state that they were able to show C/D scaRNA interaction with WRAP53 protein by utilizing a more stringent lysis condition. One interpretation of this result is that the WRAP53 protein does not directly interact with box C/D scaRNAs but instead this association is mediated by another protein factor that is liberated in the more stringent lysis conditions.

In lines 107 and 108 of the manuscript under review, Bergstrand et al. state that scaRNA2 is, for unknown reasons, more than 90% associated with chromatin (Fig. 1A). This finding

further emphasizes that scaRNA2 likely associates with other proteins that are associated with chromatin and these chromatin proteins can be part of the WRAP53 protein complex.

Reference 25 (Bergstrand, 2020): scaRNA2 is co-precipitated with GFP-WRAP53 protein (but not with point mutations of WRAP53 which cause Hoyeraal-Hreidarsson syndrome). GFP-WRAP53, but not point mutations of WRAP53, was also found to interact with proteins enriched in the Cajal body such as coilin and SMN. A direct interaction between WRAP53 protein and scaRNA2 is not shown.

Reference 37 (Venteicher, 2009): This paper does not evaluate direct binding of WRAP53 protein to scaRNAs, but utilizes IPs to determine if scaRNAs co-precipitate with the WRAP53 protein complex. Box C/D scaRNAs were not evaluated in this paper. Rather, the authors examined if WRAP53 protein associates with mixed domain scaRNAs (scaRNA10 and scaRNA5) in addition to box H/ACA scaRNAs (scaRNA8 and scaRNA13), all of which have a CAB motif.

Reference 38 (Izumikawa, 2019): The authors state that “WDR79 (WRAP53 protein) binds C/D scaRNAs several-fold less efficiently than do H/ACA scaRNAs (32,33,35), such that involvement of other unknown factor(s) in the CB localization of C/D scaRNAs has been proposed (32,33,36). In this study, we identified TDP-43 as a major player that regulates the CB localization of a subset of C/D scaRNAs independently of WDR79.”

The authors go on to state that TDP-43 is the major factor that targets scaRNA2 to Cajal bodies and WRAP53 protein interaction with scaRNA2 is dependent on TDP-43.

Response: We appreciate this thorough summary of previously published literature regarding the association between WRAP53 \$\beta\$ and scaRNA2.

2) Collectively, these and other publications strongly suggest that WRAP53 protein interaction with scaRNA2 is likely mediated by other components enriched in the Cajal body, such as TDP-43 and coilin. Hence these other proteins (TDP-43, coilin and possibly others) are probably part of the WRAP53 protein complex and facilitate a subset of box C/D scaRNA interaction, such as scaRNA2. This should be at least discussed if not tested experimentally.

Response: We are grateful for this suggestion and now demonstrate that scaRNA2 interacts directly with WRAP53 \$\beta\$, as well as with coilin, TDP-43 and fibrillarin (Fig. 7a). Co-precipitation of scaRNA2 was most pronounced in the presence of WRAP53 \$\beta\$ and coilin and these proteins also bound extensively to other scaRNAs containing either a C/D and/or H/ACA box (Extended Data Fig. 10b).

3) Until the authors formally exclude the role of TDP-43, coilin, and other CB-enriched proteins in mediating the association of scaRNA2 in the WRAP53 protein complex, the authors need to revise the manuscript text and model shown in Figure 7E. Ideally, the authors could clarify that the WRAP53 complex contains other proteins which may directly interact with box C/D scaRNAs that lack a CAB motif or GU/UG wobble stem. (The GU/UG wobble stem is another motif (Marnef, 2014) shown to interact with the WRAP53 protein). ScaRNA2 is one example of a scaRNA lacking a CAB motif and a GU/UG wobble stem and hence there is a strong possibility that other proteins in the WRAP53 complex mediate this interaction.

Response: Utilizing CLIP, scaRNA2 is now shown to interact directly with WRAP53 β (Fig. 7a) (please also see our response above).

4) Along these lines, a quick PubMed search shows that coilin has been shown to interact with Ku proteins and inhibit non-homologous DNA end joining (Velma, 2011) as well as be recruited to UVA-induced DNA lesions (Bartova, 2014). This information is very relevant to the manuscript and should be discussed. Since coilin is part of the WRAP53 complex, have the authors examined if coilin contributes to any of the proposed WRAP53 protein activities?

Response: In response to this helpful suggestion, we have now explored the potential involvement of coilin, TDP-43 and fibrillarin in the proposed activities of WRAP53 β . Immunoprecipitation of the individual components of DNA-PK revealed interaction with WRAP53 β and coilin, but not with TDP-43 or fibrillarin (Fig. 7b), indicating that a complex containing WRAP53 β and coilin is involved in activating DNA-PK.

However, in cells depleted of WRAP53 β , association between scaRNA2 and DNA-PKcs was more extensive (Fig. 7d). In contrast, this was not the case following depletion of coilin, TDP-43 or fibrillarin, which, if anything, attenuated this association (Extended Data Fig. 10e). This finding indicates that coilin does not assist WRAP53 β in regulating the interaction between scaRNA2 and DNA-PK. Future investigations will establish the functional relevance of the association between coilin and DNA-PK, as well as how this association differs from that between WRAP53 β and DNA-PK.

These data, together with the fact that coilin and TDP-43 have previously been implicated in NHEJ repair, are now included in the manuscript.

Minor issues:

1) The authors should briefly introduce box H/ACA, box C/D and mixed domain scaRNAs as well as the CAB motif, the GU/UG wobble stem and scaRNP formation and function. ScaRNA association with TDP-43 and coilin should also be discussed. It is highly likely that the scaRNA2 activity discussed in the manuscript utilizes scaRNA2 packaged into a RNP or complexed with other proteins. The authors report that 2-3% of scaRNA2 co-precipitates with DNA-PKcs (line 277). Do known box C/D scaRNP proteins such as fibrillarin also co-precipitate with DNA-PKcs?

Response: This information is now included in the Introduction. Moreover, immunoprecipitation of the individual components of DNA-PK revealed associations with WRAP53 β and coilin, but not TDP-43 or fibrillarin (Fig. 7b).

2) Line 115 states that scaRNA2 overexpression increases CB numbers (Fig. 1B), but no quantification is shown. Also, a negative control RNA overexpression is not shown to monitor for increased CB numbers as a consequence of increased transcription. In addition, no apparent increase in CB number is seen in Fig. S1F which also shows overexpressed scaRNA2. These points need to be addressed in order to support the statement in line 115.

Response: These points have now been addressed, but since effects on Cajal bodies are not the main focus of this study, we decided not include these data, but rather to remove the statement that “scaRNA2 overexpression enhances Cajal body number”. To avoid confusion,

we also replaced the image demonstrating overexpression of scaRNA2 in Fig. 1c (old Fig. 1B) with one that shows fewer Cajal bodies.

3) The scaRNA2 expression plasmid was obtained from the Kiss lab per the information in one of the supplemental tables. However, this plasmid has the pGEMT-easy backbone which does not have a mammalian promoter. Please include more details about the scaRNA2 expression vector. This information is relevant given that scaRNA2, unlike typical pol II-derived mRNAs, likely has both promoter and terminator elements that influence its expression and processing into the nucleolus-enriched mgU2-61 fragment (Tycowski et al. 2004).

Response: We now include additional details concerning the pGEMT-easy scaRNA2 plasmid in the Materials and Methods. In brief, this plasmid contains the sequence encoding full-length scaRNA2, flanked on the upstream side with 250 bp containing its own promoter and downstream with 80 bp (i.e., -250/+500) and this entire sequence was cloned into the vector utilizing EcoRI restriction sites. Since the pGEMT-easy vector lacks a mammalian promoter, expression of scaRNA2 is driven by its own promoter. The effect of the 3'-flanking sequence is currently unclear, since 3'-end processing and accumulation of scaRNA2 are governed by information contained in the mature scaRNA2 (Gérard et al., *Nucleic Acids Res* 2010).

4) Figure S1: The schematic of scaRNA2 is incorrectly annotated. According to Tycowski et al. 2004, the mgU2-61 domain consists of a C, D', C', and D box that organize into a stem-loop structure. Additionally, Izumikawa et al. (2019) found that a UG-rich motif upstream of mgU2-25 is necessary for TDP-43 binding and trafficking to the CB. The authors should edit the schematic to more accurately depict vital elements of the structure of scaRNA2.

Response: Thank you for pointing this out. This figure has now been edited accordingly (Fig. 1a and Extended Data Fig. 1a).

5) What happens to CBs and CB numbers when scaRNA2 is reduced?

Response: We have now found that in cells depleted of scaRNAs, the Cajal bodies become smaller and coilin accumulates in nucleoli. However, since, as also mentioned above, effects on Cajal bodies are not the major focus here, we decided not to include this data, although we can easily do so if the reviewer considers this to be necessary.

6) In Fig. 4, the arrangement of panel E being up by panel A is not optimal.

Response: The panels in Fig. 4 have now been rearranged.

7) In Fig. 5, have the authors tested a negative control RNA? Would the inclusion of any RNA reduce DNA-PK autophosphorylation? It is possible that the *in vitro* system might be more sensitive to any RNA whereas in cells only RNA complexes that interact with DNA-PK (such as a scaRNP2 complex) reduce autophosphorylation.

Response: scaRNA5 and scaRNA7, each of which contains a C/D domain, also diminish autophosphorylation of DNA-PK *in vitro* to varying extents (Extended Data Fig. 7b), although not all RNAs can do so (see Fig. 6i and the section below as well). We conclude that the kink-turn structure is important for the inhibition of DNA-PK catalytic activity.

8) In Fig. 7B, the WRAP53 protein was obtained by IP from U2OS cells. Consequently, proteins in the WRAP53 complex (such as TDP-43 or coilin) may be responsible for sequestering away the scaRNA2 resulting in the increase of DNA-PK autophosphorylation. The authors can easily test for the potential contribution of CB-enriched proteins in this assay by IPing WRAP53 from cells treated with siRNAs to TDP-43 or coilin or both. Alternatively, TDP-43 or coilin IP complexes can be used in this assay. For a more direct assessment of the regulation of WRAP53 in this assay, WRAP53 would need to be bacterially purified to ensure that other CB-enriched proteins are not present in the complex used for the assay.

Response: We thank the referee for this suggestion and have now performed the DNA-PK assay with WRAP53 β isolated from cells depleted of coilin, TDP-43 or fibrillarin. The findings indicate that in the absence of these proteins the ability of WRAP53 β to sequester scaRNA2 away from DNA-PKcs was unaltered (Extended Data Fig. 10c), indicating that they are not directly involved.

9) Fig. S4 D and E, lines 272-281. Based on the input signals, there is not much IP of DNA-PKcs with the 18-2 antibody. Hence it is hard to say that the epitope to which this antibody binds mediates interaction with scaRN2. A first step could be to RNase treat the lysate before 18-2 antibody IP to see if this increases DNA-PKcs recovery.

Response: We agree, this evidence is not particularly strong, but it still provides an indication, since this antibody has been used for this same purpose before. We now report that IP of the lysates of RNase A-treated cells using the 18-2 antibody increased co-precipitation of DNA-PKcs slightly (Extended Data Fig. 8h).

Response to Reviewer 2:

Bergstrand, O'Brien et al. describes the involvement of scaRNA2 in the regulation of DNA double-strand break repair pathway choice. They propose that scaRNA2 regulates DNA repair pathway choice by limiting DNA-PK activation through direct binding of scaRNA2 to the DNA-PKcs. This is proposed to stimulate Homologous Recombination, through blocking DNA-PK assembly, limiting NHEJ, promoting ATM activation and eviction of the NHEJ proteins. In the last part of the manuscript, they also involve WRAP53beta as a regulator of scaRNA2 activity: WRAP53beta is proposed to sequester scaRNA2 away from DNA-PKcs, thereby promoting efficient NHEJ. However, while the model proposed is really appealing, it is currently not based on sufficient experimental evidences. Multiple important information are missing and key aspects of the model (e.g. the direct binding of scaRNA2 to DNA-PKcs) need to be consolidated by additional experiments. Considering the large amount of experiments required to support the proposed role of scaRNA2 in DNA repair pathway choice, we feel the manuscript should be rejected.

Response: We thank the reviewer sincerely for this feedback and hope that the additional experiments now described are sufficient to confirm our conclusions.

Main comments:

- In the abstract, the authors claim that “cells lacking scaRNA2 show a pronounced reduction of the level of the MRN complex”. However, the data in Extended data Fig.S2G (transient knock-down of scaRNA2) do not support that claim. In fact the title of Extended Fig. S2 is “Loss of scaRNA2 does not alter the cell cycle or expression of repair proteins”. Consequently, the authors should remove that sentence from the abstract. In addition, they should blot for MRE11, RAD50 and NBS1 in the stable scaRNA2 conditions to complement the right panel of Extended Fig. S2G.

Response: We must apologize for the potential confusion here. The point we were trying to make was that the total levels of these factors are not affected by loss of scaRNA2, only their levels at DNA breaks. This sentence has now been rephrased to improve clarity and, moreover, Western blots for MRE11, RAD50 and NBS1 are now shown (Extended Data Fig. 4a).

- In the abstract, the authors claim that “Cells lacking scaRNA2 exhibit a pronounced reduction of (...) activation of ATM at DNA ends”. However the data do not support that claim: The effects on ATM activation as monitored after IR with PhATM and gammaH2AX are very mild: a reduction of 25% in foci intensity. The abstract need to be rephrased to convey this moderate decrease of ATM activation. The effect observed using the LacI-FokI system could be explained by reduced cutting activity of the FokI enzyme, since the control of DSB induction used here for this system is also gammaH2AX, which is itself reduced by scaRNA2 down-regulation, suggesting reduced cutting.

Response: We have now rephrased this sentence in the abstract. However, we would like to point out that activation of pATM is reduced not only at DNA breaks, but globally, as demonstrated by Western blotting (Fig. 5d).

In our opinion, the proposal that loss of scaRNA2 might reduce cutting by the FokI enzyme is unlikely, since the accumulation of total ATM, 53BP1 and ubiquitin at FokI breaks in cells lacking scaRNA2 is the same as in control cells. Moreover, in the cells lacking scaRNA2,

factors involved in NHEJ actually accumulate at FokI breaks to a greater extent than in control cells.

- What controls scaRNA2 recruitment at sites of DNA damage? The authors should deplete DNA-PKcs to test how this impact on scaRNA2 recruitment at laser lines (since based on their model, scaRNA2 associates with DNA-PKcs), and test different inhibitors (ATM, ATR, DNA-PKcs and PARP inhibitors) to determine the dependencies for the observed recruitment.

Response: This is an excellent question, which we have attempted to examine experimentally. However, visualization of scaRNA2 at DNA lesions is only possible following overexpression of this RNA and, even then, only in a fraction of the cells. This might reflect the association of a relatively small number of scaRNA2 units with each double-strand break, as is known to be the case for many NHEJ factors, and/or the transient nature of this localization. At any rate, more sensitive methods of detection are required for reliable elucidation of the mechanism by which the presence of scaRNA2 at sites of damage is controlled, as now discussed in the manuscript.

- The pictures that are shown on the side of multiple graphs do not fit with data from the quantification shown on the graphs. The authors have to show pictures with 2-3 cells instead of a single one, which is not enough to show the heterogeneous response observed. For example, showing a single cell cannot convey the decrease from 42% positive cells to 25% positive observed for RPA2 on the Fig.2A panel. Accordingly, pictures have to be replaced in Fig.2A, Fig.2B, Fig.2C, Fig.2D, Fig.3A, Fig.3B and Fig. 4B, to fit to the respective quantifications.

Response: Although the heterogeneity of the response could, indeed, be illustrated by photographs showing many cells, in our opinion the graphs quantify this situation much more clearly. As now explained in the legend to these figures, the photographs are included simply to illustrate the potential change in cell phenotype.

- In the abstract, the authors claim that “blockade of either the assembly or catalytic activity of DNA-PK in cells lacking scaRNA2 restore HR”. However, this is not what has been done in the manuscript: DNA-PK inhibition restored RAD51 loading, a step in the HR process, but not a readout of HR efficiency. The abstract should be modified accordingly. In addition, a rescue by the blockade of DNA-PK assembly was not performed (The experiment with LINP1 is indirect). This should be removed from the abstract or experiments aimed at blocking DNA-PK assembly (such as siRNA against DNA-PKcs) should be performed.

Response: In response to this helpful comment, we have re-phrased the text to make it clearer that loading of factors involved in HR was assessed. In addition, we have now performed additional experiments utilizing siRNA for DNA-PKcs to block DNA-PK assembly, which gave results similar to those obtained with siLINP1, i.e., RAD51 assembly at DNA breaks in cells lacking scaRNA2 was restored (Fig. 6c and Extended Data Fig. 8b).

- The blots in figure S3 are incomplete since in each case, the other subunits of the DNA-PK complex have to be blotted (most probably, the whole complex is present in each IP because of contaminant broken DNA in the extracts). Unless this, nothing can be concluded on the respective affinity of scaRNA2 for DNA-PKcs or Ku.

Explain why a co-IP between Ku70 and scaRNA2 is observed but not with Ku80? (Figure 4A) is the antibody interfering with scaRNA2 binding? Is the epitope of this antibody mapped? This could provide insights into the mode of interaction between Ku and scaRNA2, and suggest that scaRNA binds to Ku80 and not only to DNA-PKcs.

When an inhibition of scaRNA2 binding was observed with the 18-2 anti-DNA-PKcs antibody, this is how it was interpreted. One can propose that the antibody against Ku80 blocks the interaction with DNA-PKcs, which would explain why scaRNA2 was not found in the Ku80 IP, but then this should be tested properly by comparing the fraction of DNA-PKcs immunoprecipitated by the Ku70 and Ku80 antibodies as compared to control IP.

Response: Although this is a valid point, in most cases co-precipitation of individual subunits of the DNA-PK complex does not bring down the entire complex, probably because each subunit is present in extremely high numbers, only some of which are located in the complex. We now include additional blots in these IP figures that support this conclusion (Extended Data Figs. 6b and 7d). In fact, interaction of DNA-PKcs with Ku70/80 can only be detected following exposure to higher doses of IR, which explains why 6 Gy was employed in these co-IP experiments (Fig. 6a, b, h and Extended Data Figs. 8a). Although interaction between Ku70 and Ku80 is also detected in non-irradiated cells, a fraction of these factors remains unassociated (Extended Data Fig. 6b), which might explain why scaRNA2 is co-precipitated by Ku70, but not Ku80 under native conditions.

Nevertheless, utilizing CLIP, we now demonstrate that scaRNA2 interacts directly with DNA-PKcs, whereas the binding of Ku70/80 to scaRNA2 is primarily indirect (Fig. 4b).

Concerning the possibility that scaRNA2 blocks antibody epitopes, we found that two Ku80 antibodies that recognize different epitopes (MA5-12933 from Thermo and A302-627A from Bethyl) are both incapable of co-precipitating scaRNA2. In contrast, at least 3 different antibodies directed against DNA-PKcs co-precipitate scaRNA2, whereas the 18-2 antibody does not, indicating that scaRNA2 may, in fact, interfere with the epitope on this antibody.

- More mechanistic insights should be provided regarding the mode of interaction of scaRNA2 and DNA-PK. Is scaRNA2 binding to Ku or DNA-PKcs. This could be tested by performing RIP to check for scaRNA2 association to Ku70 in DNA-PKcs depleted cells. If the observed scaRNA2-Ku co-IP depends of DNA-PKcs this should be reduced in DNA-PKcs depleted cells.

Response: We now include data confirming that scaRNA2 interacts directly with DNA-PKcs, whereas the scaRNA2-Ku70/80 interaction is mostly indirect (Fig. 4b). Moreover, we demonstrate that scaRNA2 binds more extensively to Ku70 following depletion of DNA-PKcs from cells (Fig. 4c), indicating that DNA-PKcs and Ku70 compete for binding to scaRNA2.

- Mapping on DNA-PKcs (and Ku) the interaction domains of scaRNA2 should be performed. UV-mediated crosslink coupled with MS/MS could be used for this experiment for example.

Response: In response to this useful suggestion, the abilities of deletion variants of scaRNA2 to prevent autophosphorylation of DNA-PK *in vitro* were tested. Each C/D box domain of scaRNA2, both mgU2-25 and mgU2-61, can independently reduce this autophosphorylation and in combination this inhibition is even more pronounced (Fig. 5c). In contrast, an internal deletion mutant of scaRNA2 lacking the GU-rich region (nt 78-111) retained the ability to

inhibit DNA-PK autophosphorylation (Fig. 5c). Both the mgU2-25 and mgU2-61 motifs can adopt the kink-turn structure, indicating that this structural feature of scaRNA2 mediates the inhibition *in vitro*.

- In the *in vitro* experiments using *in vitro* synthesized scaRNA2, a control structured RNA should be used (for example TERC/scaRNA19) to confirm a specific effect, since Ku binds secondary DNA/RNA structures such as hairpins, stem loops, etc. The simplest interpretation of data in Figures 5A, B is that Ku is titrated out by binding to scaRNA2, possibly non-specifically.

Response: We now show that scaRNA7, the structure of which is similar to that of scaRNA2 (C/D motifs), also prevents autophosphorylation of DNA-PK *in vitro* (Extended Data Fig. 7b). A reduction was also obtained with scaRNA5 (containing H/ACA, C/D mixed motif), but to a lesser extent, indicating a certain degree of structural specificity with respect to this effect. In contrast, LINP1 RNA stimulated autophosphorylation of DNA-PK *in vitro* (Fig. 6i).

Inhibition of DNA-PKcs by scaRNA2 was not prevented by increasing the concentration of Ku70/80 in the reaction mixture (Extended Data Fig. 7a), providing further support for the conclusion that DNA-PKcs, not Ku70/80, is the target of scaRNA2.

- Which part of scaRNA2 is important for this activity? Various deletions should be tested (based on the predicted or known secondary structures) by complementing the scaRNA2 KO cells and monitoring its recruitment to sites of DNA damage (laser+smFISH or scaRNA2-MS2 as used in extended Fig.S2A).

Response: Our experiments designed to answer this question are described above. Moreover, as also mentioned previously, visualization of full-length scaRNA2 is challenging and even fewer FISH probes can be incorporated into the shorter variants. More sensitive methods of detection are required for reliable examination of the presence of deletion variations of scaRNA2 at DNA breaks.

- To support the model, the authors have to show that scaRNA2 depletion and overexpression affects cells survival to DNA damaging agents. From the proposed model of the implication of scaRNA2 into the DNA damage response, scaRNA2 overexpression (inhibiting NHEJ) should sensitize to IR, but not to drugs inducing DSB mainly repaired by HR such as DSBs induced by camptothecin (CPT). In contrast, scaRNA2 depletion (inhibiting HR) should sensitize to both IR and CPT. This should be tested.

Response: This important point has now been examined experimentally.

In this context, overexpression of scaRNA2 in colon cancer cells has been shown to enhance their chemoresistance (Zhang et al., J Cell Physiol 2019).

- In Fig. 5D the pattern obtained with the Phospho S2056-DNA-PKcs antibody does not seem specific (pan nuclear phosphorylation, increased in the nucleoli?), knowing the reported

frequent cross-reactivity of common Phospho S2056-DNA-PKcs antibodies (e.g. with Ph 53BP1). A control with NU7441 should be added to confirm that the staining is specific. If not, remove the figure 5D.

Response: The specificity of the pDNA-PK antibody used for immunostaining has now been confirmed by NU4771 treatment (Extended Data Fig. 7c). We would also like to point out that the strong pDNA-PK signal in the nucleolus might be due to the fact that we extracted soluble proteins prior to immunostaining in order to facilitate the visualization of proteins involved in NHEJ at DNA lesions. In this connection, components of DNA-PK present in the nucleolus have recently been reported to be resistant to extraction with detergent (Shao et al., Nature 2020).

- The authors should show the level of spontaneous DNA damage (number of gammaH2AX foci) in cells depleted (or KO) for scaRNA2 or overexpressing scaRNA2. Defective HR usually translates into increased basal DNA damage. In addition to this, scoring micronuclei could also support an increased genomic instability induced by modulation of scaRNA2 levels.

Response: Examination of spontaneous DNA damage in cells lacking scaRNA2 revealed elevated numbers of both γ H2AX foci and micronuclei, especially after the cell had been passaged a number of times (Fig. 1g, h).

- The authors should show that the decrease in NHEJ activity induced by WRAP53 depletion can be rescued by depletion of scaRNA2 to connect scaRNA2 to the observed inhibition.

Response: Co-depletion of scaRNA2 and WRAP53 β has now been performed, but rather than assessing NHEJ activity, we thought it would be more informative to examine restoration of cellular DNA-PK activation. Depletion of scaRNA2 restored hyperphosphorylation of DNA-PK in cells lacking WRAP53 β (Fig. 7e), which is consistent with scaRNA2 being responsible for DNA-PK inactivation following WRAP53 β depletion.

- The model (Fig. 7E) is confusing since it suggests that MRN binding and Ku-DNA-PKcs binding are mutually exclusive while they are not since they have a different recruitment mechanism to DSBs (MRN binding on the side of the DSB, while Ku binding at the DNA ends through accommodating DNA end in its cavity). The model should be clarified to stick to the findings: scaRNA2 inhibits DNA-PK activity and thereby promotes HR. scaRNA2 inhibits the association between Ku and DNA-PKcs (coIP Fig.6A), thereby promoting Ku eviction and HR, but it also contributes to the recruitment of CtIP and MRN (Fig.3A).

Response: We appreciate this insightful comment and have tried to emphasize the findings indicated. At the same time, we wished to address several other key findings in our model as well, including the competition between scaRNA2 and LINP1 and the involvement of WRAP53 β in this regulation. Including too many details might obscure the main message, i.e., that scaRNA2 promotes HR by blocking DNA-PK and NHEJ.

Minor comments:

- Extended Figure 1A is not clear. The same colors are used to present different features (green bar for both MS2 and smFISH probes). The MS2 is inserted into the RNA, therefore a symbol

representing this insertion would be used rather than a bar which suggests that the MS2 box is already present in the RNA.

Response: Changes made as suggested.

- In the Results section, indicate whether MCF7 and U2OS cells KO for scaRNA2 correspond to individual clones or to a population.

Response: This information is now included in the Methods.

- From the Mat & Med, it seems that the WT and scaRNA2 KO U2OS and MCF7 constitutively express Cas9. This should be indicated in the figure legend.

Response: Only the MCF7 scaRNA2 KO cells constitutively express Cas9, whereas the U2OS scaRNA2 KO cells were generated by transient transfection with Cas9. This is now clarified in the figure legend.

- Line 166-167. The accumulation of MRN and activation of ATM are not “defective” but “reduced”. Modify accordingly. In fact the reduction is quite mild.

Response: Modified accordingly.

- Figure 3 legend: again “loss of scaRNA2” does not “impairs” recruitment of MRN and ATM activation but “reduces” it. Modify accordingly.

Response: The figure legend has been modified as suggested.

- What are scaRNA2 secondary structures? (predicted or known) Considering the affinity of Ku for hairpins, this should be discussed and if relevant presented in a supplementary figure.

Response: ScaRNA2 contains two C/D domains, each of which can individually adopt the conserved kink-turn structure (Tycowski et al 2004), as now described in the manuscript. We have also found that these domains can reduce autophosphorylation of DNA-PK *in vitro* (Fig. 5c) (please see above as well).

- Is scaRNA2 transcribed by RNA polymerase II? This should be stated when describing it the first time. Is it spliced? Again this should be stated. scaRNA2 should be in bold in Table S1, since it is the focus of the study.

Response: We now explain that scaRNA2 is transcribed by RNA polymerase II as a non-spliced unit, with a 5' cap, but no poly A-tail. Moreover, scaRNA2 has been marked in bold in Supplementary Table 1.

- The use of the “in vivo” expression should be avoided since it suggests experiments in animals. For clarity, replace by “in cells”.

Response: We thank the reviewer for this specification and have now replace *in vivo* with “in cells” throughout the text.

- Extended Fig S2G, right panel: scaRNA2 KO cells show an increase in 53BP1 and BRCA1 levels. This should be discussed in the text.

Response: We thank the reviewer for this observation. Our re-assessment of the levels of 53BP1 and BRCA1 in scaRNA2 KO cells did not reveal any consistent increase. Moreover, the point we were trying to make was that the reduction of BRCA1, RAD51 and RPA2 at DNA breaks are not due to a reduction in the total levels of these proteins.

- Pladienolide B is used in Extended Fig. S2I. In the Figure legend, indicate what are the target/effects of Pladienolide B.

Response: This information is now included in the figure legend.

- Indicate in the Figure 5C and Extended data Fig. S5 legends what the star on the right corresponds to (unspecific band of which antibody?)

Response: The asterisks indicate degradation products and/or a smaller isoform of phosphorylated DNA-PKcs, as determined by MS analysis of the DNA-PK complex (Shao et al Nature 2020) and detected by all anti-pDNA-PK^{S2056} antibodies employed in our experiments. This band is most distinct following autophosphorylation of recombinant DNA-PK *in vitro*, as well as in chromatin fractions prepared from cells.

This information was already included in the legend to Fig. 5a, where blots for pDNA-PK^{S2056} are first shown. For simplicity we have now removed the asterisk from the subsequent pDNA-PK^{S2056} blots.

Response to Reviewer 3:

The manuscript by Bergstrand et al., describes how lncRNA Scar2 regulates DNA repair pathway choice by inhibiting NHEJ via DNA-PK inhibition and consequently promoting HR. This is further regulated by Scar2 binding competition between DNA-PK and WRAP53B. This story describes novel and interesting mechanism providing another evidence for RNA dependent DDR playing role not only in cis but also in trans. However, some experiments are missing key controls.

Response: Thank you for this positive overall evaluation.

Here are my comments:

Figure 1:

A: adding other RNA controls would help, for example to compare to NEAT1 and another mRNA as cytoplasmic marker

Response: In response to this helpful suggestion from the reviewer, we now include RNA controls for both the nuclear (MALAT1) and cytoplasmic (EEF2 mRNA) fractions (Extended Data Fig. 1c).

B: This experiment is missing control with scar2 GAPmer; could you add laser potency and iterations to material and methods?

Response: We now confirm the specificity of the scaRNA2 FISH staining by showing that this signal is attenuated following knockdown of scaRNA2 (Extended Data Fig. 2b). In addition, a description of the potency and iterations of the laser has been added to the Materials and Methods

C: non-specific smFISH probes should be used as negative control, in the same laser setting; 24% also seems quite low, does this mean that in 76% of damaged cells scar2 doesn't localise to the breaks? What is an explanation for this? Perhaps just transient localisation? In this experiment, scar2 is overexpressed from plasmid, could authors repeat this with endogenous levels of scar2?

Response: A probe for U2 snRNA is now included as a negative control (Fig. 1d). Unfortunately, the localization of scaRNA2 at DNA double-strand breaks can only be monitored when scaRNA2 is overexpressed and even then, only in relatively few cells. This may be due to the presence of only small numbers of scaRNA2 molecules at these lesions (as in the case of many factors involved in NHEJ) and/or the transient localization of scaRNA2 at these sites. For detailed elucidation of the mechanism regulating the presence of scaRNA2 at sites of damage, more sensitive methods of detection are required, as now discussed in the manuscript.

D: To be honest I am not convinced by these images: to me the green and red dots do not really overlap for scar2 (as they do for gH2AX) Did the authors test that adding MS2 hairpin is not affecting scar2 function/localisation; could they move it to another region on scar2, where it's not close to C and D box structures? What are the working concentration for Shield and 4OH?

Response: We agree that this overlap is not complete and now include data of overexpressed, untagged scaRNA2 that demonstrates co-localization with the FokI dots more clearly (Fig. 1e). The potential influence of MS2 on the localization of scaRNA2 is already addressed in Extended Data Fig. 2a, which demonstrates that MS2-tagged scaRNA2 localizes in the normal manner to Cajal bodies.

The working concentrations of Shield-1 and 4-OHT were each 1 μ M, as already stated in the Materials and Methods.

E: the image for scar2/3 doesn't really correspond to quantifications provided, it looks like there is more γ H2AX foci at 24 hours when compared to 1 hour time point; are they swapped?? What are the levels of other scaRNA when they use GAPmers for scar2? Can authors provide some information on off target effect? Could they show levels of scar2 after 3day post GAPmer transfection?

Response: These images, which have not been swapped, were meant to illustrate the significant increase in the number of cells containing residual γ H2AX foci following depletion of scaRNA2, i.e., the potential change in cell phenotype. At the same time, the graphs depict the heterogeneity of this response with considerably more clarity. Moreover, we did not quantify the total number of foci per cell, but only the cells containing more than 10 foci, which is our cut-off for a positive response. Thus, it is possible that, on the average, scaRNA2-depleted cells contain more foci after 72 than 48 hours of knockdown, due to the lack of repair of foci induced by IR, as well as spontaneous damage. However, this possibility was not assessed.

We now include results showing that following depletion of scaRNA2, the levels of other scaRNAs remain unchanged (Extended Data Fig. 2c) and that 72 hours after scaRNA2 depletion, the levels of this RNA was reduced by 80-90% (Extended Data Fig. 2g).

F,G: it would be interesting to know what is the phenotype for scar2 KO cells. What passage they are? In principle there could be some mutations accumulating in time, due to upregulated NHEJ and low HR. It would be also interesting to see if this effect on NHEJ sensitise them to PARPi. How in these KO cells γ H2AX looks like at 0 (no irradiation)?

Response: Examination of this interesting point revealed significant accumulation of spontaneous γ H2AX in scaRNA2 KO cells after a number of passages (Fig. 1g). Moreover, scaRNA2 KO cells displayed enhanced numbers of micronuclei compared to WT cells (Fig. 1h), which is consistent with the augmented genomic instability of scaRNA2 KO cells.

In this connection, overexpression of scaRNA2 in colon cancer cells has been shown to enhance their chemoresistance (Zhang et al., J Cell Physiol 2019).

Figure 2:

A and B: could the authors use scatter plots, comparing mean but also showing cell distribution? Also the authors should supplement these data scar2 overexpression/rescue.

Response: All of the graphs have now been converted to bar diagrams in which each replicate value and standard deviations are depicted. Moreover, the results of rescue experiments are shown in Fig. 2d and Extended Data Fig. 3a, b.

It would be great to show that lack of scar2 also affect endogenous HR sites. The authors could show less of resection in sequence specific system.

Response: Indeed, we already shown that loss of scaRNA2 impairs accumulation of RPA (resection) and RAD51/BRCA at endogenous DNA lesions following exposure to IR (Fig. 2a, b).

I am also surprised that the authors didn't see more pronounced effects on 53BP1 as one would expect based on recent study <https://www.nature.com/articles/s41467-019-12836-9>

Response: Thank you for pointing out this interesting and highly relevant article. One reason for the different findings could be that we assessed recruitment of 53BP1 into foci one hour after exposure to IR (Fig. 2a, b), while in that other study resolution of 53BP1 foci was assessed 6-72 hours after such treatment.

D: could the authors repeat this rescue with scar2 lacking MS2 overexpression.
F: the authors should add scar2-MS2 here for comparison

Response: We now include results showing that untagged scaRNA2 rescues IR-induced foci containing RAD51 and BRCA1 (Fig. 2d). Moreover, the efficiency of HR repair was enhanced following overexpression of scaRNA2 tagged with MS2 was (Extended Data Fig. 3f).

Figure 3:

C and D seems to be somewhat contradictory to Figure 1 E/F/G. Can the authors explain why in one figure they see more γ H2AX foci in cells lacking scar2, but not in another??

Response: In Figs. 1 and 3, we assessed γ H2AX at different time points, which explains the differences in the results obtained. In cells lacking scaRNA2, we observed increased numbers of cells with > 10 γ H2AX foci only 24 hours after IR irradiation and this time point was examined in Fig. 1, not in Fig. 3.

At one hour post-IR, however, the numbers of cells containing γ H2AX foci did not differ between control and scaRNA2, as seen in both Figs. 3c and 1f. In addition, in Fig. 3d we also show the intensity of γ H2AX foci one hour post-IR, which is reduced following depletion of scaRNA2.

It would good to add pATM and γ H2AX to western blots.

Response: These blots have now been added to Extended Data Fig. 4a.

B: could they test whether pATM recruitment is completely impaired or just delayed? More time after damage should be added.

Response: We now show that in cells lacking scaRNA2, the number of cells with >10 pATM foci is reduced also 6-24 hours after their exposure to IR (Extended Data Fig. 5a).

These data could be further supported by ChIP at DSBs in AsiSI for example.

Response: In the current investigation, we evaluate pATM using 3 different systems (IR-induced foci, recruitment to FokI breaks and western blotting). Application of a fourth system could be an interesting aspect of a follow-up study.

Figure 4:

A: nuclear non-specific protein control is missing

Response: We now include a RIP of nuclear MRE11 as an extra control for the antibody specificity (Extended Data Fig. 6a). Moreover, Ku80, which does not bind scaRNA2, serves as an internal control.

B: 53BP1 should added

Response: Accumulation of 53BP1 at FokI breaks is depicted in Fig. 2c.

C: the authors need to add scar2 rescue/overexpression control and KD of known NHEJ factor as a negative control

Response: We now document rescue of NHEJ by overexpression of scaRNA2 (Extended Data Fig. 6d). Concerning knockdown of a factors known to be involved in NHEJ, we consider WRAP53 β (Fig. 7f), which has been reported several times to be involved in this process (Henriksson et al., Genes Dev 2014, Hedström et al., CDDis 2015; Mata-Garrido et al., Acta Neuro Comm 2016, and Rassolzadeh et al., CDDis 2016), to constitute a valid control.

D: scar2 KD should be added for comparison

Response: In our opinion it is more logical to show the effects of scaRNA2 depletion on DNA-PK phosphorylation in the next figure (Fig. 5), where we explore the relationship between scaRNA2 and the catalytic activity of DNA-PK.

Figure 5:

A: non-specific RNA of the same length as scar2, ideally another scar should be added as a negative control

Response: We now show that scaRNA5 and scaRNA7, each of which contains a C/D domain, also caused a decline in autophosphorylation of DNA-PK *in vitro* to varying extents (Extended Data Fig. 7b). However, not all RNAs attenuate DNA-PK autophosphorylation *in vitro* (Fig. 6i).

C: here the authors use 6GY, whilst in Fig 1 they used 2 GY for irradiation. Is there any explanation for it? It would be good to be consistent.

Response: 6 Gy was applied in order to achieve more extensive interaction/co-precipitation of Ku70/80 in Fig. 6 and the findings presented in Fig. 5 were made in connection with these

same experiments. With 2 Gy, co-precipitation is barely detectable (Extended Data Fig. 7d), probably because of the high abundance of these proteins and the fact that only a small proportion of them interact. An explanation for our usage of different doses of IR is now provided in the legend to Fig. 6.

D: Scar2 overexpression/rescue should be added

Response: It is challenging to perform experiments on rescue of factors involved in NHEJ in FokI cells, since visualization of these factors requires pre-permeabilization, which strongly attenuates the FISH signal of scaRNA2. Instead, we now include data demonstrating rescue of NHEJ repair by scaRNA2 overexpression (Extended Data Fig. 6d).

E: they should add longer time point here, 1 hour or longer

Response: Our new assessment revealed that the interaction between scaRNA2 and pDNA-PK is still not restored 24 hours post-IR (Extended Data Fig. 7e).

What the NS t-test corresponds to? Surely not to no IR and 5min. Is it shifted?

Response: This graph has been replaced (Fig. 5f).

Figure S3:

A and B Nuclear protein controls are missing

Response: We already include IgG as a negative control and it is unclear to us what additional nuclear protein the reviewer would like us to probe. We also use GAPDH as a control, since the IP was performed on whole-cell lysate. In any case, we now include blots for the nuclear proteins Ku70 and Ku80 in Extended Data Figs. 6b and 7d.

Figure 6:

A, B and C: nuclear protein control is missing

Response: As also explained in our response above, we included GAPDH as a negative control, since the IP was performed on whole-cell lysate. Unspecific IgG was not tested, since this experiment was designed to determine the difference between Control and scaRNA2#1-treated samples.

What are the levels of LINP on chromatin after Scar2 depletion?

Response: There were no changes in the expression of LINP1 or its localization to chromatin following depletion of scaRNA2 (Extended Data Fig. 9h) or generation of DNA damage (Extended Data Fig. 9g).

Furthermore, the authors show that scar2 prevents DNA-PK autophosphorylation and complex assembly. They also show that there is a competition between scar2 and LINP1 lncRNA, where sca2 is a negative regulator, whilst LINP2 is a positive one. It would be interesting to understand what determines the winner of this competition. Is expression of these lncRNAs cell cycle regulated, are they antagonistically expressed? If not, in which situation DNA-PK would preferentially bind to either scar2 or LINP1? Could author extend their studies to provide more clarifications for this?

Response: These are interesting questions, which we have now addressed. During the various phases of the cell cycle the level of scaRNA2 remained relatively constant, with somewhat higher expression during the G2 phase, when repair by HR is favored (Fig. 6j). In contrast, expression of LINP1 was lowest during the G2 and highest during the G1 phase (Fig. 6j), when NHEJ is dominant.

Applying IP, we found that depletion of LINP1 elevated the binding of scaRNA2 to pDNA-PKcs (Fig. 6g), while depletion of scaRNA2 enhanced binding of LINP1 to pDNA-PK (Fig. 6f and Extended Data Fig. 9e), confirming that scaRNA2 and LINP1 compete for binding to DNA-PKcs.

Finally, LINP1 RNA stimulated autophosphorylation of DNA-PK (Fig. 6i), whereas DNA-PK remained inactive in the presence of both scaRNA2 and LINP1 (Fig. 6i). Thus, inhibition of DNA-PK by scaRNA2 appears to be more potent than stimulation by LINP1.

Figure 7:

S4D and E: these data should be supported with in vitro studies

Response: We have now confirmed that DNA and RNA reduce precipitation of recombinant DNA-PKcs by the 18-2 antibody *in vitro* (Extended Data Fig. 8g). Moreover, we demonstrate that the ability of this antibody to precipitate DNA-PKcs in cells depleted of RNA by RNase A is enhanced slightly (Extended Data Fig. 8h).

B: could add LINP here?

Response: This question has been explored as one aspect of Fig. 6i (please see our response above).

Similarly, the authors show that WRAP53B is in competition with DNA-PK for scaRNA2 binding. Again, how is this regulated? Any changes depending on cell cycle phase or perhaps in response to DNA damage? It would be great to understand this molecular mechanism a bit more.

Response: In response to these helpful comments, we now provide additional data concerning the potential involvement of the Cajal body proteins coilin, TDP-43 and fibrillarin in this regulation, as well as the possibility that these proteins influence the ability of WRAP53 β to sequester scaRNA2 away from DNA-PK (Fig. 7 and Extended Data Fig. 10).

Does WRAP35B bind to LINP1?

Response: No binding between WRAP53 β and LINP1 was detected in U2OS cells (Fig. 7a).

Does WRAP35B bind to DNA-PK?

Response: Immunoprecipitation of the individual components of DNA-PK revealed interaction with WRAP53 β (Fig. 7b).

[Redacted]

[Redacted]

[Redacted]

[Redacted]

REVIEWER COMMENTS

Reviewer #1 (Remarks to the Author):

The authors have done an outstanding job in addressing my previous concerns.

Reviewer #2 (Remarks to the Author):

The revised version addressed some, but not all, of my comments but also raise some new concerns that need to be addressed through a novel round of revisions:

1. As now mentioned by the authors “Visualization of scaRNA2 at DNA lesions was only possible following overexpression of this RNA and, even then, only in a fraction of the cells.” This is worrisome. This sentence should also appear in the Results section when presenting the Fig.1d and Fig.1e. The percentage of cells showing scaRNA2 a sites of DNA damage should also appear in the text of the manuscript (not just on the pictures).

2. The authors would not analyze the dependencies for the association of scaRNA2 to sites of DNA damage, stating “This is an excellent question, which we have attempted to examine experimentally. However, visualization of scaRNA2 at DNA lesions is only possible following overexpression this RNA and, even then, only in a fraction of the cells.” However, in Fig.1D, they have 25% of cells showing scRNA2 at DNA damage sites induced by laser microirradiation, while and in Fig.1E, they have 31% of cells showing scaRNA2 recruitment at FokI-induced DNA damage. These numbers are high enough to test different inhibitors (ATM, ATR, DNA-PKcs and PARP inhibitors) and provide a mechanism for this association. To my view, this experiment should and can be performed.

[REDACTED]

Minor comments:

- Line 198-199 “the MRN complex recruits and activates ATM 199 (through phosphorylation at S1981)...” should be clarified. It sounds like MRN is a kinase phosphorylating ATM.

Reviewer #3 (Remarks to the Author):

The authors now provide much better manuscript. They addressed all my comments and provided requested controls. These new data now strengthen the claims and conclusions of the manuscript. Furthermore, I find very interesting that the authors see cell cycle and LINP1/scaRNA2 potential competition correlation.

This study now provides important contribution to DDR field. The roles of RNA in trans at DSBs are not well understood and might be very relevant to overall control of DNA damage repair.

Point-by-point response to the reviewers' comments

Response to Reviewer 1

The authors have done an outstanding job in addressing my previous concerns.

Response: We thank the reviewer for this positive feedback.

Response to Reviewer 2:

The revised version addressed some, but not all, of my comments but also raise some new concerns that need to be addressed through a novel round of revisions:

1. As now mentioned by the authors “Visualization of scaRNA2 at DNA lesions was only possible following overexpression of this RNA and, even then, only in a fraction of the cells.” This is worrisome. This sentence should also appear in the Results section when presenting the Fig.1d and Fig.1e. The percentage of cells showing scaRNA2 a sites of DNA damage should also appear in the text of the manuscript (not just on the pictures).

Response: This information is now included in the Results.

2. The authors would not analyze the dependencies for the association of scaRNA2 to sites of DNA damage, stating “This is an excellent question, which we have attempted to examine experimentally. However, visualization of scaRNA2 at DNA lesions is only possible following overexpression this RNA and, even then, only in a fraction of the cells.” However, in Fig.1D, they have 25% of cells showing scRNA2 at DNA damage sites induced by laser microirradiation, while and in Fig.1E, they have 31% of cells showing scaRNA2 recruitment at FokI-induced DNA damage. These numbers are high enough to test different inhibitors (ATM, ATR, DNA-PKcs and PARP inhibitors) and provide a mechanism for this association. To my view, this experiment should and can be performed.

Response: Our main goal in this connection was simply to demonstrate the presence of scaRNA2 at DNA breaks. However, the signal associated with scaRNA2 is in general weak, particularly at DNA lesions, where the laser micro-irradiation procedure itself contributes to variation between experiments, due to inherent difficulties with calibration and focus. Consequently, we do not feel that we can provide reliable detailed characterization of scaRNA2 accumulation at sites of DNA damage. If this reviewer considers this situation to be problematic, we would be willing to remove our quantification of scaRNA2 at laser stripes/FokI breaks and only show the images, along with an explanation that the presence of scaRNA2 at break sites is observed in only a fraction of the cells.

In this context it is important to keep in mind that factors involved in NHEJ are in general difficult to detect at double-strand breaks. The presence of several of these at such sites has only recently been visualized employing high-resolution imaging following pretreatment with RNase and detergent.

[REDACTED]

[REDACTED]

[REDACTED]

[REDACTED]

Minor comments:

- Line 198-199 “the MRN complex recruits and activates ATM 199 (through phosphorylation at S1981)...” should be clarified. It sounds like MRN is a kinase phosphorylating ATM.

Response: This is now clarified.

Response to Reviewer 3:

The authors now provide much better manuscript. They addressed all my comments and provided requested controls. These new data now strengthen the claims and conclusions of the manuscript. Furthermore, I find very interesting that the authors see cell cycle and LINP1/scaRNA2 potential competition correlation. This study now provides important contribution to DDR field. The roles of RNA in trans at DSBs are not well understood and might be very relevant to overall control of DNA damage repair.

Response: We thank the reviewer for this positive feedback.

[Redacted]

[Redacted]

[Redacted]

[Redacted]